# Improving Reward Models with Proximal Policy Exploration for Preference-Based Reinforcement Learning

**Yiwen Zhu**[1,2*]   **Jinyi Liu**[3*]   **Pengjie Gu**[2*]   **Yifu Yuan**[3]   **Zhenxing Ge**[4]
**Wenya Wei**[1]   **Zhou Fang**[1†]   **Yujing Hu**[5]   **Bo An**[2,6]
[1]Zhejiang University   [2]Nanyang Technological University   [3]Tianjin University
[4]Nanjing University   [5]NetEase Fuxi AI Lab   [6]Skywork AI

## Abstract

Reinforcement learning (RL) heavily depends on well-designed reward functions, which are often biased and difficult to design for complex behaviors. Preference-based RL (PbRL) addresses this by learning reward models from human feedback, but its practicality is constrained by a critical dilemma: while existing methods reduce human effort through query optimization, they neglect the preference buffer's restricted coverage — a factor that fundamentally determines the reliability of reward model. We systematically demonstrate this limitation creates distributional mismatch: reward models trained on static buffers reliably assess in-distribution trajectories but falter with out-of-distribution (OOD) trajectories from policy exploration. Crucially, such failures in policy-proximal regions directly misguide iterative policy updates. To address this, we propose Proximal Policy Exploration (PPE) with two key components: (1) a *proximal-policy extension* method that expands exploration in undersampled policy-proximal regions, and (2) a *mixture distribution query* method that balances in-distribution and OOD trajectory sampling. By enhancing buffer coverage while preserving evaluation accuracy in policy-proximal regions, PPE enables more reliable policy updates. Experiments across continuous control tasks demonstrate that PPE enhances preference feedback utilization efficiency and RL sample efficiency over baselines, highlighting preference buffer coverage management's vital role in PbRL.

## 1   Introduction

In reinforcement learning (RL), the reward function is pivotal as it specifies the learning objectives and guides agents toward desired behaviors. Traditional RL has made significant achievements in complex domains such as gaming and robotics, largely due to well-designed reward functions [1, 2, 3]. Yet, constructing these functions presents significant challenges. Designing suitable reward functions that accurately encapsulate complex behaviors like cooking or summarizing books is time-consuming and prone to human cognitive biases [4, 5, 6, 7, 8]. Additionally, embedding social norms into these functions remains unresolved [9].

An emerging alternative that addresses some of these challenges is preference-based reinforcement learning (PbRL). This approach bypasses the need for meticulously engineered rewards by leveraging overseer preferences between pairs of agent behaviors, which is typically gathered from humans [10, 11, 12, 13, 14, 15, 16, 17]. In PbRL, agents learn to optimize behaviors that align with

---

*Equal contribution

†Corresponding author: zfang@zju.edu.cn

Code is available at: `https://github.com/yiwenzhu-evan/PPE`

the demonstrated human preferences, offering a more intuitive and flexible method for performing desired behaviors.

Despite its advantages, PbRL typically requires extensive preference feedback, which can be labor-intensive, time-consuming and sometimes infeasible to gather, potentially limiting its applicability in real-world settings where rapid adaptation is essential [13, 14, 15]. To overcome these challenges, prior research has explored various strategies for improving feedback efficiency. These strategies include selecting the most informative queries to improve the quality of the learned reward function while minimizing the required teacher input [12, 18, 19, 20]. Also, techniques such as sampling based on ensemble disagreements, mutual information, or behavior entropy have been employed to target behaviors to refine the overall reward model more effectively [10, 13, 16, 18, 20]. Moreover, QPA [21] ensures that queries and policy learning progress concurrently, significantly reducing feedback unrelated to the current policy and enhancing feedback efficiency.

Central to these methods is the so-called "preference buffer", a dataset that stores pairs of trajectory segments along with human preference labels. Analogous to the replay buffer in standard RL, the preference buffer collects and maintains the preference data that the reward model is trained on. While the term "preference buffer" is not yet universally standardized, managing and sampling from such a dataset is fundamental to almost all modern PbRL algorithms [10, 13].

However, these methods overlook the relationship between the preference buffer and the effectiveness of the reward model. Without understanding this relationship, the reward model may not generalize well to data outside the distribution of the preference buffer. This can lead the reward model to inaccurately evaluate out-of-distribution data, potentially resulting in misguided policy improvements.

To address this issue, we focus on enhancing the coverage of the preference buffer. Basically, our findings revealed that the learned reward model provides more precise evaluations for trajectories that fall within the preference buffer's distribution. This insight led us to develop the Proximal Policy Exploration (PPE) algorithm. Firstly, we need to train an out-of-distribution (OOD) detection mechanism to evaluate whether newly encountered data from the environment falls outside the preference buffer's distribution. Using the OOD degree measurement of the current data, we employ the *proximal-policy extension* method, which encourages the agent to explore data that, while beyond the preference buffer's distribution, still aligns closely with the current policy. Furthermore, we have designed the *mixture distribution query* method, which not only actively queries data outside the preference buffer's distribution but also queries a portion of the in-distribution data. The aim of this approach is to actively expand the preference buffer's coverage while avoiding a reduction in the reward model's evaluation accuracy for in-distribution trajectories due to insufficient volume of in-distribution data. By integrating these two methods, we are able to broaden the preference buffer's coverage and bolster the reliability of the reward model's evaluations for the near-policy distribution.

Overall, our work systematically addresses a key bottleneck in preference-based reinforcement learning: the limited generalization of the reward model due to insufficient coverage of the preference buffer. By explicitly expanding the buffer's distribution and ensuring reliable reward evaluation, our approach enables more robust and efficient policy learning in practical settings.

Our main contributions are as follows:

1. We introduce an OOD detection mechanism to ascertain whether data falls outside the preference buffer's distribution, and formulate the behavior policy resolution as a constrained optimization problem for exploring such data.

2. For this constrained optimization problem, we provide a closed-form approximation. Through this, we introduce the *proximal-policy extension* method in PPE, an analytical behavior policy that directly explores data outside the preference buffer's distribution. This approach actively enhances the coverage of the preference buffer.

3. We have found that the reliability of the reward model is heavily dependent on the data distribution; the reward model can only provide reliable assessments when there is sufficient data within the evaluated distribution. To address this, we propose a *mixture distribution query* method in PPE, which balances the volume of in-distribution and out-of-distribution query data, ensuring accurate evaluations by the reward model across different regions.

4. Extensive experiments across multiple benchmark tasks demonstrate that PPE significantly improves both exploration efficiency and final policy performance compared to existing preference-based RL methods, validating the effectiveness and generality of our approach.

## 2 Preliminaries

**Preference-based RL**   In PbRL, an agent that interacts with an environment in discrete time steps. At each time step $t$, the agent at state $s_t$ selects an action $a_t$ based on its policy. Unlike traditional RL, where the environment returns a reward $r(s_t, a_t)$ evaluating the agent's behavior, PbRL employs preference feedback. Here, a teacher provides preferences between pairs of agent behaviors, which uses to learn proxy rewards that align with human preferences, guiding policy adjustment [10, 11, 12, 22, 23].

Formally, a behavior segment $\tau$ consists of a sequence of time-indexed observations and actions $\{(s_t, a_t), \ldots, (s_{t+H}, a_{t+H})\}$. Given a pair of segments $(\tau^0, \tau^1)$, the teacher gives their preference feedback signal $y_p$ among these segments, identifying preferred behaviors or marking segments as equally preferred or incomparable. The primary objective in PbRL is to train the agent to perform behaviors aligned with human with minimal feedback.

The PbRL learning process involves two main steps: (1) *Agent Learning*: The agent interacts with the environment to collect experiences and updates its policy using existing RL algorithms to maximize the sum of proxy rewards. (2) *Reward Learning*: The reward model $\hat{r}_\psi$ is optimized based on feedback received from the teacher, denoted as $(\tau^0, \tau^1, y_p) \sim \mathcal{D}^p$. This cyclical process continually refines both the policy and the reward model, detailed in Appendix A.

**OOD Detection**   Neural networks are known for making confident predictions, even when encountering out-of-distribution (OOD) samples [24, 25, 26]. A common approach for OOD detection involves fitting a generative model to the dataset, which assigns high probability to in-distribution samples and low probability to OOD ones. Although effective for simple, unimodal data, these methods can become computationally intensive when dealing with more complex and multimodal data. An alternative approach trains classifiers to act as more sophisticated OOD detectors [27].

In this study, we focus on Morse neural networks [28], which train a generative model to produce an unnormalized density that equals to 1 at the dataset modes. We utilize this model to generate a metric that assesses the extent to which current data deviates from the preference buffer distribution. A Morse neural network produces an unnormalized density $M(x) \in [0, 1]$ on an embedding space $\mathbb{R}^e$, attaining a value of 1 at mode submanifolds and decreasing towards 0 when moving away from the mode [28]. The rate at which the value decreases is controlled by a Morse Kernel. More details about the Morse neural network can be found in Appendix B.

**Preference buffer**   We formally define the preference buffer, $\mathcal{D}^p$, as the dataset containing all collected preference tuples, denoted as $\mathcal{D}^p = \{(\tau_i^0, \tau_i^1, y_{p,i})\}_{i=1}^N$. Each tuple consists of a pair of trajectory segments $(\tau^0, \tau^1)$ and a preference label $y_p$ (e.g., $y_p \in \{0, 1\}$), indicating the preferred segment. The reward model, $\hat{r}_\psi$, is trained exclusively on data sampled from this buffer. A core premise of our work is that the distribution of trajectories within $\mathcal{D}^p$ is critical for the reward model's generalization. Therefore, our proposed methods are explicitly designed to monitor and strategically expand the coverage of this buffer.

In this paper we use "preference buffer" ($\mathcal{D}^p$) as the dataset of labeled preference tuples ($\mathcal{D}^p = (\tau_0, \tau_1, y_p)$), where $(\tau_0, \tau_1)$ are trajectory (or segment) snippets and $y_p \in \{0, 1\}$ indicates which one is preferred. The reward model is trained exclusively on samples drawn from ($\mathcal{D}^p$), and our query policy/selection schemes explicitly aim to expand the coverage of ($\mathcal{D}^p$) in policy-proximal regions.

## 3 Method

In this chapter, we delve into the importance of preference buffer coverage for the reward model in our study and discuss strategies to actively expand this coverage.

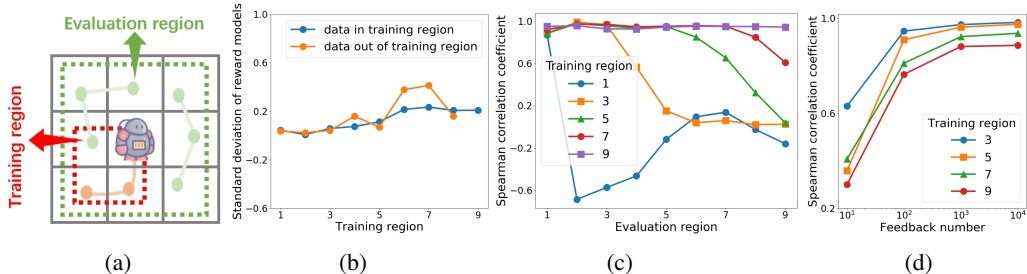

(a)          (b)          (c)          (d)

Figure 1: Observe the reward model's effectiveness in a random walk task with a grid world. **(a)** Training the reward model with preference data generated from trajectory pairs within the training region marked by the red frame, and assessing the correlation between the proxy and ground truth returns across all trajectories in the evaluation region denoted by the green frame; **(b)** The variance in the proxy rewards associated with transitions inside and outside of the training region changes in the size of the training region; **(c)** The Spearman correlation coefficient between proxy returns and ground truth returns for all trajectories in various evaluation regions, using reward models trained with preference data from different training regions; **(d)** The Spearman correlation coefficient varies with the number of feedbacks used to train the reward model in different training regions.

## 3.1   Why Coverage is Important? — A Motivating Example

We designed an experiment to observe the relationship between the effectiveness of the reward model and the coverage of transitions in the preference buffer used to train the reward model.

As shown in Figure 1, we set up an environment in a grid world where the robot can move in four directions: up, down, left, and right. Each cell in the grid world has an associated ground truth reward, and the ground-truth return for a given trajectory is calculated by summing the predefined, ground-truth rewards of each cell visited in that trajectory. It should be noted that Figure 1(a) serves as a schematic representation; in reality, the grid world is structured as a 9x9 grid. Additionally, the horizontal axes in Figures 1(b) and 1(c) represent the side lengths of the respective region, while the horizontal axis in Figure 1(d) represents the number of feedbacks. Specifically, 'region X' refers to a square area with the lower-left corner at (0,0), bounded by the segments from 0 to X on both axes. For example, 'region 3' denotes the grid area from (0,0) to (3,3).

We designated two areas within the grid world as the training and evaluation regions, as illustrated in Figure 1(a). First, we uniformly sampled 1,000 trajectory pairs of length 3 in the training region via a random walk policy (i.e., uniformly sampling actions). We sampled with replacement, so duplicate trajectories are possible, which is acceptable as the experiment focuses on the spatial coverage of training data rather than the diversity of trajectories within that region. Based on the relative sizes of their ground truth returns, we assigned preference labels to these pairs and stored them in a preference buffer. Next, we trained a reward model using data from the preference buffer with a Bradley-Terry loss. Finally, we evaluated all trajectories of length 6 in the evaluation region using the learned reward model. The goal of this evaluation is not to assess a specific policy, but rather to assess the reward model's ability to correctly rank these pre-existing trajectories. To further analyze the effectiveness of the reward model, we assessed the correlation between the proxy returns computed by the reward model and the ground truth returns using the Spearman correlation coefficient.

**Preference Buffer Coverage Should Be an Explicit Optimization Objective:** Results in Figure 1(c) show that expanding the training region significantly enhances the reward model's ability to accurately evaluate trajectories. For instance, the blue curve in Figure 1(c), representing a tiny 1x1 training region, shows a sharp decrease in performance as the evaluation region expands. It starts with a high Spearman correlation when evaluated on the same 1x1 region but plummets as the evaluation region grows to include states the reward model has never seen. The slight rebound of Spearman correlation when the evaluation region becomes very large does not indicate genuine generalization; it is a statistical artifact of rank correlation. As the region expands, the pool includes more obviously high- vs. low-return trajectories, which the model can still order correctly despite widespread OOD errors, nudging the correlation upward from a strongly negative regime toward weak correlation. This result strongly supports our central claim that reward models generalize poorly to out-of-distribution

trajectories. This finding highlights the critical importance of increasing the coverage of the preference buffer over the transition space. If the preference buffer does not comprehensively cover the transition distribution of the current policy, the proxy rewards generated by the reward model may be unreliable, making policy optimization ineffective. Only with extensive buffer coverage can the reward model reliably evaluate a broader range of trajectories. Given these observations, it is essential to explicitly include preference buffer coverage as an optimization objective in the PbRL pipeline.

**Ensemble Variance Cannot Identify Out-of-Distribution Transitions:** To create the ensemble baseline, we trained multiple reward models (5 in our experiment) on the exact same preference buffer. Diversity was introduced only through different random network initializations and mini-batch shuffling during training, a common technique for uncertainty estimation. As demonstrated in Figure1(b), the variance in outputs from ensemble reward models, given the same transition input, does not distinguish whether the transition belongs to the training region. Therefore, RUNE[15] cannot actively expand the preference buffer's coverage.

**Balancing In-Distribution and Out-of-Distribution Feedback is Crucial:** To explore how varying the amount of preference feedback affects performance, we designed the experiment in Figure 1(d) such that the evaluation region is the same as the training region. This setup allows us to isolate the impact of feedback quantity while keeping the regions fixed. Figure 1(d) shows that, within the same training region, increasing the amount of feedback improves the evaluation accuracy of the reward model. This indicates that it is not sufficient to focus solely on collecting out-of-distribution data; it is also necessary to ensure that new queries include enough in-distribution data. Maintaining this balance is crucial to prevent the reward model from inaccurately evaluating regions it has already explored.

Consequently, to train a reliable reward model, it is essential not only for the agent to actively explore OOD data to expand the preference buffer coverage but also to ensure that there is a sufficient amount of in-distribution data within the preference buffer.

## 3.2 How to Expand Coverage of Preference Buffer? — Proximal Policy Exploration

Based on the observations in Section 3.1, we propose the PPE algorithm, which includes two core modules: the *proximal-policy extension* method to enhance preference buffer coverage, and the *mixture distribution query* method to balance the inclusion of in-distribution and out-of-distribution data. By leveraging transition uncertainty estimation, PPE combines these methods to develop a more reliable reward model within the current policy distribution.

**Leveraging Morse Neural Network for Transition Uncertainty Estimation**  Drawing inspiration from the work of [29], we propose $f_\phi$ as a perturbation model that generates an action $\hat{a} = f_\phi(s, a)$. This implies that $\hat{a} = a$ only when the pair $(s, a)$ originates from the preference buffer $\mathcal{D}^p$. Simultaneously, the preference buffer $\mathcal{D}^p$ is composed of tuples $(\tau^0, \tau^1, y_p)$, where each segment $\tau$ is a sequence of state-action pairs $\{(s_t, a_t), \ldots, (s_{t+H}, a_{t+H})\}$.

$$M_\phi(s, a) = 1 - K_{RBF}(f_\phi(s, a), a), \quad \text{where} \quad K_{RBF}(z_1, z_2) = e^{-\frac{\lambda^2}{2}\|z_1 - z_2\|^2}. \tag{1}$$

Based on this, we design the Morse Neural Network such that $M_\phi(s_i, a_j) = 0$ is valid only when $\{s_i, a_j\} \in \mathcal{D}^p$. In particular, we utilize a Radial Basis Function (RBF) kernel [30] to shape the Morse Network, as illustrated in Eq.(1).

Subsequently, we optimize this Morse Neural Network by minimizing the KL divergence between unnormalized measures [31], as detailed in [28]. This can be expressed as $D_{KL}(\mathcal{D}^p(s, a) \| 1 - M_\phi(s, a))$. Hence, in terms of $\phi$, this implies minimizing the loss depicted in Eq.(2). Additional details can be found in Appendix C.

$$L(\phi) = \frac{1}{N} \sum_{s,a \sim \mathcal{D}^p} \left[ \frac{\lambda^2}{2} \|f_\phi(s, a) - a\|^2 + \frac{1}{M} \sum_{a_u \sim \text{Uniform}(\mathcal{A})} \exp^{-\frac{\lambda^2}{2}\|f_\phi(s, a_u) - a_u\|^2} \right]. \tag{2}$$

Here, $a_u$ signifies an action sampled from a uniform distribution over the corresponding action space, denoted as $\text{Uniform}(\mathcal{A})$. Furthermore, $M$ represents the number of samples drawn from $\text{Uniform}(\mathcal{A})$, while $N$ refers to the number of sampled $(s, a)$ pairs from $\mathcal{D}^p$. The parameter $\lambda$ is used to control the sensitivity of the Morse Neural Network to OOD transitions.

**Expanding Preference Buffer Coverage via Proximal-Policy Extension Method** Observations from Figure 1(c) suggest that expanding the coverage of the preference buffer can enhance the ability of the trained reward model in evaluating the quality of trajectories. Particularly during the RL training process, only when the trained reward model has a strong ability to evaluate the quality of trajectories within the proximal policy distribution can the risk of misguidance in policy improvement be reduced. Therefore, expanding the coverage of the preference buffer for the proximal policy distribution can further optimize policy improvement in PbRL.

Drawing on this insight, we have designed the *proximal-policy extension* method, to actively encourage the agent to explore data that falls outside the preference buffer distribution but within the vicinity of the current policy's distribution. The behavior policy $\pi_E$ used for exploration, is designed such that the state-action pairs $(s, a)$ it generates when interacting with the environment can support the distribution produced by the current target policy $\pi_T$. Formally, the behavior policy $\pi_E = \mathcal{N}(\mu_E, \Sigma_E)$ is defined as the solution to the constrained optimization problem in Eq.(3).

$$\max_{\mu, \Sigma} \mathbb{E}_{a \sim \mathcal{N}(\mu, \Sigma)} [M_\phi(s, a)],$$
$$\text{s.t. } D_{KL}(\mathcal{N}(\mu, \Sigma) | \mathcal{N}(\mu_T, \Sigma_T)) \leq \epsilon. \tag{3}$$

Since we need to calculate the constrained optimization problem described in Eq.(3) in each interaction process, using readily available solvers would result in a significant consumption of computational resources. Therefore, we tighten the constraint conditions to obtain a closed-form approximate solution as shown in Proposition 3.1. This approach greatly reduces the computational cost of solving the constrained optimization problem, while achieving our desired objective of encouraging exploration of data out of the preference buffer distribution near the current policy distribution. The detailed derivation is presented in Appendix D.

**Proposition 3.1.** *The behavior policy for exploration resulting from Eq.(3) has the form* $\pi_E = \mathcal{N}(\mu_E, \Sigma_E)$, *where*

$$\mu_E = \mu_T + \frac{\sqrt{2\epsilon} \cdot \Sigma_T [\nabla_a M_\phi(s, a)]_{a=\mu_T}}{\sqrt{[\nabla_a M_\phi(s, a)]_{a=\mu_T}^T \Sigma_T [\nabla_a M_\phi(s, a)]_{a=\mu_T}}}, \ \Sigma_E = \Sigma_T. \tag{4}$$

**Mixture Distribution Query Selection** In the previous section, we introduced an exploration method that enables the agent to explore a broader range of transitions that are out of the preference buffer but near the current policy distribution. These newly discovered transitions are stored in the replay buffer. Therefore, it becomes essential to have a query selection method that can select those segments that are out of the preference buffer and store them in the preference buffer.

---

**Algorithm 1** *Mixture Distribution Query*

**Input:** $\tau \in \mathcal{D}^{cp}$, $M_\phi(\tau)$, query size $b$ and mixture ratio $\kappa$.
**Output:** $\{\tau^0, \tau^1\}_{i=1}^b$
1 **for** $i = 1$ *to* $\kappa b$ **do**
2    $\lfloor \ \tau^0, \tau^1 \sim P^{out}(\tau)$                            // Sample $\tau$ outside the distribution of $\mathcal{D}^p$
3 **for** $i = 1$ *to* $(1 - \kappa) b$ **do**
4    $\lfloor \ \tau^0, \tau^1 \sim P^{in}(\tau)$                               // Sample $\tau$ inside the distribution of $\mathcal{D}^p$
5 **return** $\{\tau^0, \tau^1\}_{i=1}^b$

---

Additionally, as inspired by the phenomenon demonstrated in Figure 1(d), if we merely select those segments outside the preference buffer's distribution and store them in the preference buffer, it implies that the volume of data in the in-distribution region will not undergo substantial expansion. As a result, the evaluation capability of the trained reward model in the in-distribution region may become less reliable due to the lack of sufficient data in this area.

Taking all these factors into account, we propose the *mixture distribution query* method. This method aims to actively select out-of-distribution data to increase the preference buffer coverage, while also selecting some in-distribution data for query. This method not only proactively increases the

coverage of the preference buffer but also boosts the volume of in-distribution data, thereby ensuring the evaluation capability of the reward model in the in-distribution region.

Specifically, for all $\tau \in \mathcal{D}^{cp}$, we can express the degree of a segment of trajectory $\tau$ being out of the preference buffer distribution according to Eq.(5), where $\mathcal{D}^{cp}$ represents the data to be queried. A higher value indicates that the data is more likely to be in-distribution.

$$M_\phi(\tau) = \frac{1}{|\tau|} \sum_{(s,a) \in \tau} M_\phi(s,a). \tag{5}$$

The size of $\mathcal{D}^{cp}$ is not large, typically $|\mathcal{D}^{cp}| \ll |\mathcal{D}|$, especially when combined with the *policy-aligned query* technique proposed in QPA [21], the quantity of $\mathcal{D}^{cp}$ is further reduced. Under this premise, we can redistribute the sampling probability for $\tau \in \mathcal{D}^{cp}$.

As shown in Eq.(6), we designed two probability density functions $P^{in}(\cdot)$ and $P^{out}(\cdot)$ according to the degree of in-distribution and out-of-distribution, respectively representing the probability of sampling $\tau$ according to the degree of in-distribution and out-of-distribution. We use a mixture ratio $\kappa \in [0, 1]$ to control the proportion of samples drawn from each distribution. A larger $\kappa$ indicates a higher proportion of samples are drawn from $P^{out}(\cdot)$.

$$P^{in}(\tau) = \frac{1 - M_\phi(\tau)}{\sum_{\tau' \in \mathcal{D}^{cp}} [1 - M_\phi(\tau')]}, \; P^{out}(\tau) = \frac{M_\phi(\tau)}{\sum_{\tau' \in \mathcal{D}^{cp}} M_\phi(\tau')}. \tag{6}$$

It's worth noting that, as mentioned in the preceding paragraph, we need to calculate $M_\phi(s,a)$ for each newly encountered $(s,a)$ when using *proximal-policy extension* method. Therefore, by maintaining $\{M_\phi(s,a)|(s,a) \in \mathcal{D}^{cp}\}$ and updating it regularly, we can avoid recalculating $M_\phi(s,a)$ when using the *mixture distribution query*, thus saving a significant amount of overhead. The specific procedure is illustrated in Algorithm 1.

**Proximal Policy Exploration Algorithm** In summary, the *proximal-policy extension* method and the *mixture distribution query* method complement each other. The use of the *mixture distribution query* method can mitigate potential issues that might arise from solely using the *proximal-policy extension* method. The combination of these two methods forms our PPE algorithm, with the algorithmic process detailed in Algorithm 2.

---

**Algorithm 2** Proximal Policy Exploration

**Input:** Query frequency $K$, feedback size per query $b$, mixture ratio $\kappa$ and Morse buffer $\mathcal{D}^m$
1  Unsupervised pretraining
2  **for** *each iteration* **do**
3      $a \sim \pi_E(\cdot|s)$                    // Sample action via *proximal-policy extension*, Eq.(4)
4      $\{s, a, M_\phi(s,a)\} \cup \mathcal{D}^m$                  // Store the OOD metric of transition
5      Store new transition $(s,a)$
6      **if** *iteration%K == 0* **then**
7         $\left. \begin{array}{l} \{\tau^0, \tau^1\}_{i=1}^{(1-\kappa)b} \sim P^{in}(\tau) \\ \{\tau^0, \tau^1\}_{i=\kappa b+1}^{\kappa b} \sim P^{out}(\tau) \end{array} \right\}$     // *Mixture distribution query*, Algorithm 1
8         Query for preference $\{y\}_{i=1}^b$
9         Store preference $\mathcal{D}^p \leftarrow \mathcal{D}^p \cup \{\tau^0, \tau^1, y\}_{i=1}^b$
10        **for** *each gradient step* **do**
11           $\mathcal{B} \leftarrow \{\tau^0, \tau^1, y\}_{i=1}^h \sim \mathcal{D}^p$        // Sample a minibatch preference data
12           Training the reward model
13           Optimize loss of $M_\phi$ in Eq.(2) *w.r.t.* $\phi$ using $\mathcal{B}$
14        Relabel the reward in $\mathcal{D}$
15        Relabel the OOD metric via $M_\phi(\cdot)$ for $(s,a) \in \mathcal{D}^{cp}$
16     **for** *each gradient step* **do**
17        Optimize $\pi_T$ via SAC method

---

In Algorithm 2, $\mathcal{D}^m = \{(s, a, M_\phi(s,a))|(s,a) \in \mathcal{D}^{cp}\}$. The parts highlighted in brown represent the additions made by our algorithm compared to the basic algorithm framework.

Improvements in different algorithms typically focus on various stages: the data storage stage (Line 5, QPA [21]), the data selection for querying stage (Line 7, QPA, B-Pref [21, 12]), the reward model update stage (Line 12, SURF, PEBBLE [14, 13]), and the agent update stage (Line 17, RUNE, QPA [15, 21]). Our approach, however, primarily enhances the data exploration stage, offering the advantage of excellent compatibility with existing methods. Although we use the *mixture distribution query* method for data selection, it does not conflict with existing query methods. We can apply the *mixture distribution query* method as a post-processing step on the results of existing query methods to select suitable data for querying. This further demonstrates the compatibility of our approach.

In practical applications, PPE can be implemented as an algorithmic plugin within an existing framework. This integration enhances the policy exploration process without requiring extensive modifications to the current framework.

## 4 Experiments

Our method, as outlined in Section 3.2, is designed to be orthogonal and highly compatible with existing strategies. Notably, our *mixed distributed query* technique does not interfere with the *policy alignment query* employed in the QPA method. This compatibility allows us to seamlessly integrate PPE into the QPA algorithm for subsequent experiments. To simplify our discussion, we will directly refer to this integrated approach as PPE henceforth.

We conducted an evaluation of our method using the MetaWorld [32] and DMControl [33] benchmarks. For a comprehensive comparison, we selected several baselines, including PEBBLE [13], SURF [14], RUNE [15], and the previous state-of-the-art method, QPA [21]. We conducted experiments using five different random seeds. For each task, we record the mean and standard deviation of the final 10 evaluation episodes aggregated across all random seeds. For a complete understanding of our experimental details, please refer to Appendix H. Moreover, we also made use of the official code repositories provided in the papers of the corresponding baseline algorithms for a fair comparison.

### 4.1 Benchmark Task Performance

Table 1: Performance comparison across benchmark tasks

| Task | PEBBLE | SURF | RUNE | QPA | PPE |
|---|---|---|---|---|---|
| Walker-walk | $453.43 \pm 159.43$ | $661.01 \pm 91.72$ | $414.62 \pm 182.16$ | $796.08 \pm 147.94$ | $\mathbf{908.09 \pm 55.30}$ |
| Walker-run | $188.21 \pm 79.86$ | $237.65 \pm 116.85$ | $251.48 \pm 104.98$ | $416.52 \pm 222.01$ | $\mathbf{520.18 \pm 101.72}$ |
| Quadruped-walk | $364.46 \pm 138.98$ | $554.20 \pm 279.01$ | $506.89 \pm 266.33$ | $621.51 \pm 259.35$ | $\mathbf{660.71 \pm 179.67}$ |
| Quadruped-run | $319.35 \pm 118.78$ | $285.51 \pm 114.13$ | $231.88 \pm 59.57$ | $386.90 \pm 118.41$ | $\mathbf{431.32 \pm 115.87}$ |
| Cheetah-run | $545.77 \pm 130.00$ | $556.78 \pm 59.323$ | $508.60 \pm 186.06$ | $578.89 \pm 133.14$ | $\mathbf{644.91 \pm 30.37}$ |
| Humanoid-stand | $249.48 \pm 218.74$ | $306.60 \pm 167.76$ | $279.45 \pm 220.00$ | $453.65 \pm 23.75$ | $\mathbf{577.12 \pm 30.93}$ |
| Drawer-open | $20.00 \pm 44.72$ | $40.09 \pm 54.89$ | $67.20 \pm 44.21$ | $40.09 \pm 54.89$ | $\mathbf{79.61 \pm 44.55}$ |
| Sweep-into | $75.45 \pm 43.11$ | $57.81 \pm 53.09$ | $\mathbf{99.62 \pm 0.56}$ | $80.67 \pm 27.00$ | $96.47 \pm 8.47$ |
| Hammer | $41.31 \pm 53.57$ | $85.23 \pm 26.18$ | $91.86 \pm 17.77$ | $78.75 \pm 44.04$ | $\mathbf{96.27 \pm 5.19}$ |
| Door-open | $72.42 \pm 35.82$ | $91.47 \pm 11.04$ | $98.23 \pm 1.92$ | $\mathbf{100.00 \pm 0.00}$ | $\mathbf{100.00 \pm 0.00}$ |
| Door-unlock | $99.65 \pm 0.55$ | $99.60 \pm 0.55$ | $\mathbf{100.00 \pm 0.00}$ | $98.43 \pm 1.14$ | $\mathbf{100.00 \pm 0.00}$ |
| Button-press | $76.69 \pm 6.02$ | $78.81 \pm 14.38$ | $88.66 \pm 7.13$ | $72.28 \pm 29.81$ | $\mathbf{90.06 \pm 5.43}$ |
| Window-open | $87.14 \pm 17.08$ | $87.29 \pm 26.99$ | $96.77 \pm 4.14$ | $90.15 \pm 20.09$ | $\mathbf{100.00 \pm 0.00}$ |

**Locomotion Tasks in DMControl Suite:** We evaluate PPE on six locomotion tasks from DMControl: *Walker-walk*, *Walker-run*, *Cheetah-run*, *Humanoid-stand*, *Quadruped-walk*, and *Quadruped-run*.

**Robotic Manipulation Tasks in MetaWorld:** We also test on challenging MetaWorld manipulation tasks: *Hammer*, *Sweep-into*, *Drawer-open*, *Door-open*, *Door-unlock*, *Button-press*, and *Window-open*. Following previous work [10, 12, 14, 15, 21], we use the ground truth success rate as the evaluation metric.

Unless otherwise noted, we fix the mixture ratio ($\kappa = 0.5$) and the KL constraint ($\epsilon = 0.01$) for all tasks and environments, with no per-task tuning.

As shown in Table 1, PPE consistently achieves superior performance across all benchmark tasks, where bold values indicate a statistically significant improvement over the second-best method . This indicates that PPE can more effectively select and utilize feedback under limited query budgets. This supports our claim that expanding the coverage of the preference buffer improves the reward model's evaluation capability and leads to more reliable policy updates.

Furthermore, our results show that PPE greatly improves feedback efficiency across complex tasks. While RUNE [15] underperforms on DMControl tasks, it achieves the second-best performance after PPE on MetaWorld tasks. We also find that the performance variance of PbRL algorithms is higher in MetaWorld than in DMControl, consistent with previous studies [12, 14, 21, 15].

## 4.2 Ablation Study

To further investigate the impact of each component in PPE, we conducted additional ablation experiments on the Walker-walk task. These experiments aim to provide empirical evidence for the parameter selection of PPE.

To assess the roles of the *proximal-policy extension* method (EXT) and the *mixture distribution query* method (MDQ) within PPE, we incrementally applied these methods to the backbone algorithm QPA. As shown in Figure 2(a), using either EXT or MDQ alone does not result in significant improvements. As described in Section 3.2, EXT and MDQ complement each other. Using only EXT increases the amount of out-of-preference buffer distribution data in the replay buffer without directly enhancing the coverage of the preference buffer. Conversely, using only MDQ fails to introduce sufficient out-of-preference buffer distribution data into the preference buffer due to the lack of active exploration, thus not effectively strengthening the reward model. Therefore, the superior performance of PPE arises from the mutual compensation of the shortcomings of EXT and MDQ.

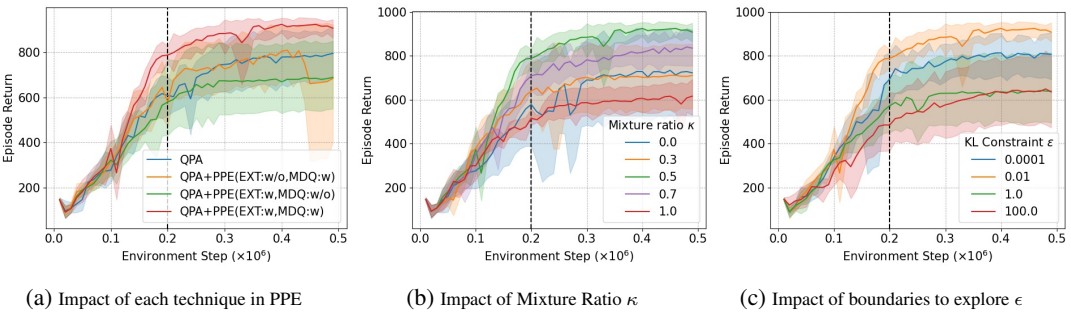

(a) Impact of each technique in PPE  (b) Impact of Mixture Ratio $\kappa$  (c) Impact of boundaries to explore $\epsilon$

Figure 2: Various ablation studies on the Walker-walk task. The dashed black line indicates the final feedback collection step. Shaded areas indicate standard deviation; the solid line shows the mean.

Next, we examine the effect of the mixture ratio $\kappa$ in MDQ with complete PPE, which determines the balance of in-distribution and out-of-distribution data in the preference buffer. As Figure 2(b) shows, optimal performance is achieved at $\kappa = 0.5$. This result confirms that indiscriminate addition of out-of-distribution data to the preference buffer can overextend distribution boundaries, undermining the reward model's effectiveness as it relies on both the coverage of the preference buffer and balanced data density across trajectory regions. Furthermore, exclusive in-distribution data sampling could hinder the reward model's adaptability to new distributions of trajectories after policy updates.

Additionally, we conducted ablation experiments on the KL constraint $\epsilon$ in Eq.(4). This parameter represents the exploration boundary of EXT for out-of-distribution data. As discussed in Appendix D, theoretically, using EXT requires that the behavior policy and target policy do not differ significantly. This implies that if $\epsilon$ is too large, performance cannot be theoretically guaranteed. Conversely, if $\epsilon$ is too small, EXT loses its exploratory significance. Experimental results, shown in Figure 2(c), confirm this property: both excessively large and small values of $\epsilon$ negatively impact the results. Therefore, we recommend setting $\epsilon$ to 0.01.

# 5 Related Work

**Human-in-the-loop Reinforcement Learning:** Human-in-the-loop RL leverages human preferences, typically collected via comparative judgments, to guide agent behavior [34, 10, 35]. The main challenge lies in the high cost and efficiency of preference acquisition [13, 15, 14]. Existing methods range from online preference modeling [36] to recursive optimization [4].

**Query Selection in PbRL:** Efficient query selection is crucial for improving feedback efficiency in PbRL. Traditional approaches use entropy-based metrics, feature distances, and spatial sampling strategies such as K-medoids and Poisson disk sampling [18, 20, 37, 38]. Recent work highlights *query-policy misalignment* as a key limitation, where selected queries may not be the most informative for improving the current policy [21]. The Query-Policy Alignment (QPA) method [21] addresses this by selecting queries expected to maximally impact the policy. Our proposed PPE method complements this perspective. While QPA focuses on the immediate utility of a query for policy optimization, PPE aims to improve the long-term quality of the preference buffer by encouraging exploration of diverse, underrepresented state-action regions. Thus, PPE can be integrated with frameworks like QPA, where PPE first guides the agent to discover novel behaviors, from which QPA can then select the most policy-relevant queries.

**Exploration in RL:** Balancing exploration and exploitation is fundamental in RL [22]. In the context of PbRL, methods like RUNE [15] primarily drive exploration using disagreement among an ensemble of reward models. The rationale is that high disagreement signals regions of high uncertainty about the true reward function, which are valuable to query. However, this uncertainty-based exploration can be inefficient, as it may repeatedly explore regions that are difficult to model but not necessarily helpful for discovering new, high-reward behaviors. In contrast, our approach focuses on maximizing preference buffer coverage. This encourages the agent to visit novel state-action regions that are underrepresented in the preference data, thereby improving the global quality and robustness of the learned reward model, rather than just focusing on areas of high model uncertainty. More broadly, RL exploration strategies include uncertainty-driven, intrinsic motivation, and information-theoretic methods [39, 40, 41, 42, 43].

# 6 Conclusion and Discussion

This paper highlights the critical role of preference buffer coverage in the evaluative accuracy of reward models. Our findings indicate that a reward model's accuracy is the highest for trajectories within the preference buffer's distribution and significantly decreases for out-of-distribution trajectories. We introduce the PPE algorithm, which actively expands the preference buffer coverage to enhance the reliability of the reward model, comprising two complementary components: the *proximal-policy extension* method and the *mixture distribution query* method. These components synergistically work to expand the preference buffer coverage while balancing the inclusion of both in-distribution and out-of-distribution data. PPE provides a more reliable reward model, thereby reducing the potential of misleading policy improvements. PPE has demonstrated substantial gains in feedback and sample efficiency through extensive evaluations on the DMControl and MetaWorld benchmarks. These results underscore the importance of actively expanding preference buffer coverage in PbRL research.

In this study, our main focus is on enhancing the reward model's quality by actively expanding the preference buffer's coverage. However, our current query method does not consider the variations in information between different pairs of agent behaviors. As we advance our research, we plan to investigate advanced methods to boost feedback efficiency. We believe that considering factors such as data similarity and clustering traits can further refine and optimize our query method.

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

# A   The Process of Reward Model Training in PbRL

Using a preference dataset $\mathcal{D}^p$, the reward model $\hat{r}_\psi$ learns to assign higher proxy returns $\hat{G}_\psi = \sum_t \hat{r}_\psi(s_t, a_t)$ to preferred trajectories. Employing the Bradley-Terry model [44], the probability that one trajectory is preferred over another is computed as:

$$P_\psi(\tau^1 \succ \tau^0) = \frac{\exp\left(\sum_t \hat{r}_\psi(s_t^1, a_t^1)\right)}{\sum_{i \in \{0,1\}} \exp\left(\sum_t \hat{r}_\psi(s_t^i, a_t^i)\right)}. \tag{7}$$

The probability estimate $P_\psi$ is used to minimize the cross-entropy between the predicted and true preference labels:

$$L_{CE} = -\mathbb{E}_{(\tau^0, \tau^1, y_p) \sim \mathcal{D}^p} \left[ \mathbb{I}\{y_p = (\tau^0 \succ \tau^1)\} \log P_\psi(\tau^0 \succ \tau^1) + \mathbb{I}\{y_p = (\tau^1 \succ \tau^0)\} \log P_\psi(\tau^1 \succ \tau^0) \right]. \tag{8}$$

After optimizing the reward function $\hat{r}_\psi$ from human preferences, PbRL algorithms enable training of RL agents with standard RL algorithms, treating the proxy rewards from $\hat{r}_\psi$ as if they were ground truth rewards from the environment.

# B   Information About Morse Neural Network

**Definition B.1** (Morse Kernel). A Morse Kernel is a positive definite kernel $K$. When applied in a space $Z = \mathbb{R}^k$, the kernel $K(z_1, z_2)$ takes values in the interval $[0, 1]$ and satisfies $K(z_1, z_2) = 1$ if and only if $z_1 = z_2$.

All kernels of the form $K(z_1, z_2) = e^{-D(z_1, z_2)}$, where $D(\cdot, \cdot)$ is a divergence [31], are considered Morse Kernels. In this study, we utilize the Radial Basis Function (RBF) Kernel,

$$K_{RBF}(z_1, z_2) = e^{-\frac{\lambda^2}{2} \|z_1 - z_2\|^2}, \tag{9}$$

where $\lambda$ is a scale parameter of the kernel [30].

Consider a neural network that maps from a feature space $X$ to a latent space $Z$ via a function $f_\phi : X \to Z$, with parameters $\phi$. Here, $X \in \mathbb{R}^d$ and $Z \in \mathbb{R}^k$. A Morse Kernel can be used to impose structure on the latent space.

**Definition B.2** (Morse Neural Network). A Morse neural network is defined as a function $f_\phi : X \to Z$ combined with a Morse Kernel $K(z, t)$, where $z \subset Z$ is a target chosen as a hyperparameter of the model. The Morse neural network is expressed as $M_\phi(x) = 1 - K(f_\phi(x), t)$.

According to Definition B.1, $M_\phi(x)$ takes values in the interval $[0, 1]$. When $M_\phi(x) = 0$, $x$ corresponds to a mode that aligns with the level set of the submanifold of the Morse neural network. Additionally, $1 - M_\phi(x)$ represents the certainty that the sample $x$ is from the training dataset, making $M_\phi(x)$ a measure of the epistemic uncertainty of $x$.

The function $-\log[1 - M_\phi(x)]$ quantifies a squared distance, $d(\cdot, \cdot)$, between $f_\phi(x)$ and the nearest mode in the latent space at $m$:

$$d(z) = \min_{m \in M} d(z, m), \tag{10}$$

where $M$ is the set of all modes. This provides information about the topology of the submanifold and satisfies the Morse–Bott non-degeneracy condition [45].

The Morse neural network exhibits the following properties:

1. $M_\phi(x) \in [0, 1]$;
2. $M_\phi(x) = 0$ at its mode submanifolds;
3. $-\log[1 - M_\phi(x)] \geq 0$ represents a squared distance that satisfies the Morse–Bott non-degeneracy condition on the mode submanifolds;
4. Since $M_\phi(x)$ is an exponentiated squared distance, the function is distance-aware, meaning that as $f_\phi(x) \to t$, $[1 - M_\phi(x)] \to 1$.

## C  Derivation of The Loss Function For Morse Neural Network In PbRL

We achieve the measurement of whether the current data is outside the distribution of $\mathcal{D}^p$ using the Morse Neural Network by minimizing the KL divergence $D_{KL}(\mathcal{D}^p(s,a)\|1-M_\phi(s,a))$. The detailed derivation process is as follows:

$$\min_\phi \mathop{\mathbb{E}}_{s,a\sim\mathcal{D}^p}\left[\log\frac{\mathcal{D}^p(s,a)}{1-M_\phi(s,a)}\right] + \mathop{\mathbb{E}}_{s\sim\mathcal{D}^p}\left[\frac{1}{|\mathcal{A}|}\int_{a\in\mathcal{A}}1-M_\phi(s,a)-\mathcal{D}^p(s,a)da\right].$$

$$\rightarrow\min_\phi \mathop{\mathbb{E}}_{s,a\sim\mathcal{D}^p}\left[-\log\left[1-M_\phi(s,a)\right]+\mathop{\mathbb{E}}_{a_u\sim\mathrm{Uniform}(\mathcal{A})}\left[1-M_\phi(s,a)\right]\right].$$

$$\rightarrow\min_\phi\frac{1}{N}\sum_{s,a\sim\mathcal{D}^p}\left[-\log K_{RBF}(f_\phi(s,a),a)+\frac{1}{M}\sum_{a_u\sim\mathrm{Uniform}(\mathcal{A})}K_{RBF}(f_\phi(s,a_u),a_u)\right]. \tag{11}$$

$$\rightarrow\min_\phi\frac{1}{N}\sum_{s,a\sim\mathcal{D}^p}\left[\frac{\lambda^2}{2}\|f_\phi(s,a)-a\|^2+\frac{1}{M}\sum_{a_u\sim\mathrm{Uniform}(\mathcal{A})}\exp^{-\frac{\lambda^2}{2}\|f_\phi(s,a_u)-a_u\|^2}\right].$$

## D  Proof of Propostion 3.1

Consider the formula for the KL divergence between two high-dimensional Gaussian distributions:

$$D_{KL}(\mathcal{N}(\mu,\Sigma),\mathcal{N}(\mu_T,\Sigma_T))=\frac{1}{2}\left[(\mu-\mu_T)^\mathrm{T}\Sigma_T^{-1}(\mu-\mu_T)-\log\det(\Sigma_T^{-1}\Sigma)+tr(\Sigma_T^{-1}\Sigma)-n\right]. \tag{12}$$

When $D_{KL}(\mathcal{N}(\mu,\Sigma),\mathcal{N}(\mu_T,\Sigma_T))\leq\epsilon$ is employed as a constraint, the solution to the optimization problem $\underset{\mu,\Sigma}{argmax}\,\mathbb{E}_{a\sim\mathcal{N}(\mu,\Sigma)}[M_\phi(s,a)]$ is typically achieved through iterative means. However, considering our objective for the calculated $\mu,\Sigma$ to more effectively explore data from the out-of-preference buffer distribution within the proximal policy region, and the real-time requirement for problem-solving with each agent-environment interaction, we propose a more efficient closed-form approximation to the original problem by appropriately tightening the constraint, as shown in Proposition 3.1.

We introducing $\Sigma=\Sigma_T$, and the tightened constraint can be expressed as:

$$D_{KL}(\mathcal{N}(\mu,\Sigma_T),\mathcal{N}(\mu_T,\Sigma_T))\leq\epsilon.$$

$$\rightarrow\frac{1}{2}\left[(\mu-\mu_T)^\mathrm{T}\Sigma_T^{-1}(\mu-\mu_T)-\log\det(\Sigma_T^{-1}\Sigma_T)+tr(\Sigma_T^{-1}\Sigma_T)-n\right]\leq\epsilon. \tag{13}$$

$$\rightarrow\frac{1}{2}\left[(\mu-\mu_T)^\mathrm{T}\Sigma_T^{-1}(\mu-\mu_T)\right]\leq\epsilon.$$

Substituting this into Eq.(3), we derive a simplified optimization problem:

$$\max_\mu \mathop{\mathbb{E}}_{a\sim\mathcal{N}(\mu,\Sigma_T)}[M_\phi(s,a)],$$

$$\mathrm{s.t.}(\mu-\mu_T)^\mathrm{T}\Sigma_T^{-1}(\mu-\mu_T)\leq2\epsilon. \tag{14}$$

To address the problem in Eq.(14), we construct the following Lagrangian function:

$$L=M_\phi(s,a)-\xi((\mu-\mu_T)^\mathrm{T}\Sigma_T^{-1}(\mu-\mu_T)-2\epsilon). \tag{15}$$

Deriving with respect to $\mu$ yields:

$$\nabla_\mu L=\nabla_a M_\phi(s,a)|_{a=\mu}-\xi\Sigma_T^{-1}(\mu-\mu_T). \tag{16}$$

Setting $\nabla_\mu L=0$, we find:

$$\mu=\mu_T+\frac{1}{\xi}\Sigma_T\,\nabla_a M_\phi(s,a)|_{a=\mu}. \tag{17}$$

By applying the KKT conditions, we deduce:

$$(\mu - \mu_T)^\mathrm{T} \Sigma_T^{-1} (\mu - \mu_T) - 2\epsilon = 0.$$
$$\xi > 0. \tag{18}$$

Further, via plugging Eq.(17) in Eq.(18), we can solve to obtain:

$$\frac{1}{\xi^2} \left( \Sigma_T \left. \nabla_a M_\phi(s,a) \right|_{a=\mu} \right)^\mathrm{T} \Sigma_T^{-1} \left( \Sigma_T \left. \nabla_a M_\phi(s,a) \right|_{a=\mu} \right) = 2\epsilon, \ \xi > 0.$$

$$\rightarrow \xi^2 = \frac{[\nabla_a M_\phi(s,a)]_{a=\mu}^\mathrm{T} \Sigma_T [\nabla_a M_\phi(s,a)]_{a=\mu}}{2\epsilon}, \ \xi > 0. \tag{19}$$

$$\rightarrow \xi = \sqrt{\frac{[\nabla_a M_\phi(s,a)]_{a=\mu}^\mathrm{T} \Sigma_T [\nabla_a M_\phi(s,a)]_{a=\mu}}{2\epsilon}}.$$

Through Eq.(19), we find that $\xi$ is a function of $\mu$. However, Eq.(17) is a differential equation, which is challenging to solve directly for $\mu$. Therefore, we perform a Taylor expansion on $[\nabla_a M_\phi(s,a)]_{a=\mu}$:

$$\left. \nabla_a M_\phi(s,a) \right|_{a=\mu} \approx \left. \nabla_a M_\phi(s,a) \right|_{a=\mu_T} + \left. \nabla_a^2 M_\phi(s,a) \right|_{a=\mu_T} (\mu - \mu_T). \tag{20}$$

This implies that when $\mu$ is sufficiently close to $\mu_T$, we can approximate:

$$\left. \nabla_a M_\phi(s,a) \right|_{a=\mu} \approx \left. \nabla_a M_\phi(s,a) \right|_{a=\mu_T}. \tag{21}$$

Since our goal is to increase the density of proximal policy data in the preference buffer, thereby enhancing the reward model's evaluation capability under the current policy distribution, this approximation does not conflict with our objective and is indeed very fitting.

Thus, further, we can deduce:

$$\mu \approx \mu_T + \frac{\sqrt{2\epsilon} \cdot \Sigma_T [\nabla_a M_\phi(s,a)]_{a=\mu_T}}{\sqrt{[\nabla_a M_\phi(s,a)]_{a=\mu_T}^\mathrm{T} \Sigma_T [\nabla_a M_\phi(s,a)]_{a=\mu_T}}}. \tag{22}$$

Therefore, the exploration behavior policy $\mathcal{N}(\mu_E, \Sigma_E)$ can be expressed as

$$\mu_E = \mu_T + \frac{\sqrt{2\epsilon} \cdot \Sigma_T [\nabla_a M_\phi(s,a)]_{a=\mu_T}}{\sqrt{[\nabla_a M_\phi(s,a)]_{a=\mu_T}^\mathrm{T} \Sigma_T [\nabla_a M_\phi(s,a)]_{a=\mu_T}}}, \text{and } \Sigma_E = \Sigma_T. \tag{23}$$

# E  Additional Experiments

## E.1  The p-values for the Welch's T-test

Table 2: Welch's t-test p-values for performance comparisons between PPE and other baselines throughout all training evaluations

| Task \Algorithm | PEBBLE | SURF | RUNE | QPA |
|---|---|---|---|---|
| Walker-walk | $< 0.001$ | $< 0.001$ | $< 0.001$ | $< 0.001$ |
| Walker-run | $< 0.001$ | $< 0.001$ | $< 0.001$ | $< 0.001$ |
| Quadruped-walk | $< 0.001$ | $< 0.001$ | $< 0.001$ | $0.2977$ |
| Quadruped-run | $< 0.001$ | $< 0.001$ | $< 0.001$ | $< 0.001$ |
| Cheetah-run | $< 0.001$ | $< 0.001$ | $< 0.001$ | $< 0.001$ |
| Humanoid-stand | $< 0.001$ | $< 0.001$ | $< 0.001$ | $< 0.001$ |
| Drawer-open | $< 0.001$ | $0.0412$ | $0.1675$ | $0.0078$ |
| Hammer | $< 0.001$ | $< 0.001$ | $0.3041$ | $0.0160$ |
| Sweep-into | $< 0.001$ | $< 0.001$ | $0.0184$ | $< 0.001$ |

To facilitate the intuitive demonstration of the performance advantages of the PPE algorithm in our main text experiments, we have also tabulated the p-values of the Welch's T-test comparing the performance of the PPE algorithm with that of other baselines in all evaluations during training in Table 2.

For most tasks, the algorithms PEBBLE, SURF, RUNE, and QPA show significant differences from PPE (p-value $< 0.05$), indicating that these algorithms perform significantly differently from PPE on these tasks.

In three cases, the p-value is greater than 0.05, such as for RUNE in the Drawer-open task, RUNE in the Hammer task, and QPA in the Quadruped-walk task. Although there is no significant difference, PPE remains one of the best methods.

## E.2 Exploration Methods Across Different Backbones

As shown in Tables 3 and 4, we applied PPE and RUNE to QPA and PEBBLE, respectively. This approach not only verifies the compatibility of PPE but also highlights the performance differences of various exploration methods across different backbones.

Table 3: The Performance of Different Exploration Methods on PEBBLE

| Task \Algorithm | PEBBLE | PEBBLE+RUNE | PEBBLE+PPE |
|---|---|---|---|
| Walker-walk | $453.43 \pm 159.43$ | $414.62 \pm 182.16$ | $\mathbf{499.73 \pm 82.75}$ |
| Walker-run | $188.21 \pm 79.86$ | $251.48 \pm 104.98$ | $\mathbf{257.64 \pm 58.59}$ |
| Quadruped-walk | $364.46 \pm 138.98$ | $\mathbf{506.89 \pm 266.33}$ | $451.06 \pm 223.27$ |
| Quadruped-run | $319.35 \pm 118.78$ | $231.88 \pm 59.57$ | $\mathbf{373.09 \pm 149.10}$ |
| Cheetah-run | $545.77 \pm 130.00$ | $508.60 \pm 186.06$ | $\mathbf{569.54 \pm 84.27}$ |
| Humanoid-stand | $249.48 \pm 218.74$ | $279.45 \pm 220.00$ | $\mathbf{357.13 \pm 76.15}$ |

Table 4: The Performance of Different Exploration Methods on QPA

| Task \Algorithm | QPA | QPA+RUNE | QPA+PPE |
|---|---|---|---|
| Walker-walk | $796.08 \pm 147.94$ | $704.39 \pm 133.45$ | $\mathbf{908.09 \pm 55.30}$ |
| Walker-run | $416.52 \pm 222.01$ | $429.66 \pm 173.62$ | $\mathbf{520.18 \pm 101.72}$ |
| Quadruped-walk | $621.51 \pm 259.35$ | $593.61 \pm 295.84$ | $\mathbf{660.07 \pm 175.58}$ |
| Quadruped-run | $386.90 \pm 118.41$ | $367.71 \pm 108.01$ | $\mathbf{433.42 \pm 116.58}$ |
| Cheetah-run | $578.89 \pm 133.14$ | $\mathbf{689.52 \pm 49.39}$ | $644.91 \pm 30.37$ |
| Humanoid-stand | $453.65 \pm 23.75$ | $419.74 \pm 27.38$ | $\mathbf{577.12 \pm 30.93}$ |

## E.3 Ablation Study on $\kappa$

Table 5: Impact of Mixture Ratio $\kappa$ on Walker-walk performance with 100 feedback instances

| $\kappa$ | Episode Return | $\kappa$ | Episode Return | $\kappa$ | Episode Return |
|---|---|---|---|---|---|
| 0.0 | $722.33 \pm 256.97$ | 0.4 | $756.41 \pm 215.27$ | 0.8 | $696.62 \pm 243.53$ |
| 0.1 | $795.26 \pm 174.23$ | 0.5 | $\mathbf{908.09 \pm 55.30}$ | 0.9 | $744.50 \pm 173.46$ |
| 0.2 | $688.53 \pm 212.85$ | 0.6 | $714.03 \pm 230.37$ | 1.0 | $616.22 \pm 106.39$ |
| 0.3 | $710.22 \pm 187.74$ | 0.7 | $834.91 \pm 103.28$ | | |

Under the Walker-walk experiment setting with 100 feedback instances, we investigated the impact of the mixture ratio $\kappa$ on the experimental results, as shown in Table 5. Based on these results, we set the mixture ratio $\kappa$ to 0.5 for all subsequent experiments.

## E.4 Ablation Study on the Various Components of PPE

We denote the *proximal-policy extension* method as EXT and the *mixture distribution query* method as MDQ. The specific details are recorded in Table 6.

Table 6: Performance Impact of PPE Components on Walker-Walk Task (100 Feedback Instances)

| Algorithm | Episode Return | Algorithm | Episode Return |
|---|---|---|---|
| QPA | $796.08 \pm 147.94$ | QPA+PPE(EXP:w,MDQ:w) | $\mathbf{908.09 \pm 55.30}$ |
| QPA+PPE(EXP:w,MDQ:w/o) | $689.33 \pm 194.50$ | QPA+PPE(EXP:w/o,MDQ:w) | $685.03 \pm 346.27$ |

## E.5 Ablation Study on KL Constraint $\epsilon$

In Table 7, we present the impact of different KL constraints $\epsilon$ on the performance

Table 7: Impact of various KL Constraint $\epsilon$ on Walker-walk performance (100 Feedback Instances)

| KL Constraint $\epsilon$ | Episode Return | KL Constraint $\epsilon$ | Episode Return |
|---|---|---|---|
| 1e-4 | $806.67 \pm 137.13$ | 1e-2 | $\mathbf{908.09 \pm 55.30}$ |
| 1e-1 | $745.54 \pm 163.10$ | 1e0 | $638.53 \pm 202.73$ |
| 1e1 | $783.26 \pm 163.70$ | 1e2 | $635.46 \pm 214.01$ |

## E.6 Ablation study on more environments

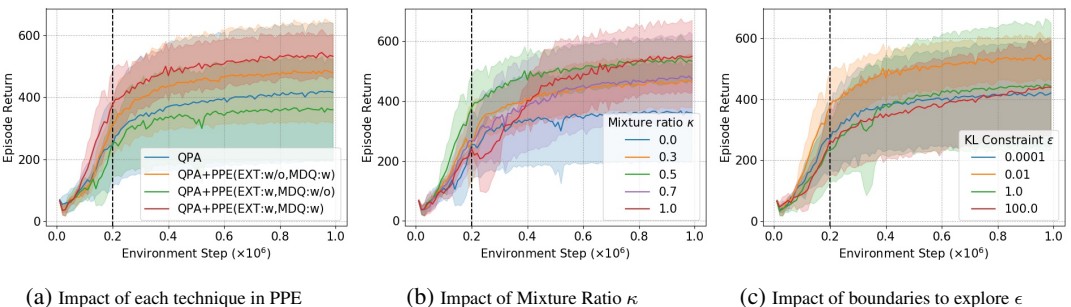

(a) Impact of each technique in PPE   (b) Impact of Mixture Ratio $\kappa$   (c) Impact of boundaries to explore $\epsilon$

Figure 3: Various ablation studies on the Walker-run task. The dashed black line indicates the final feedback collection step. Shaded areas indicate standard deviation; the solid line shows the mean.

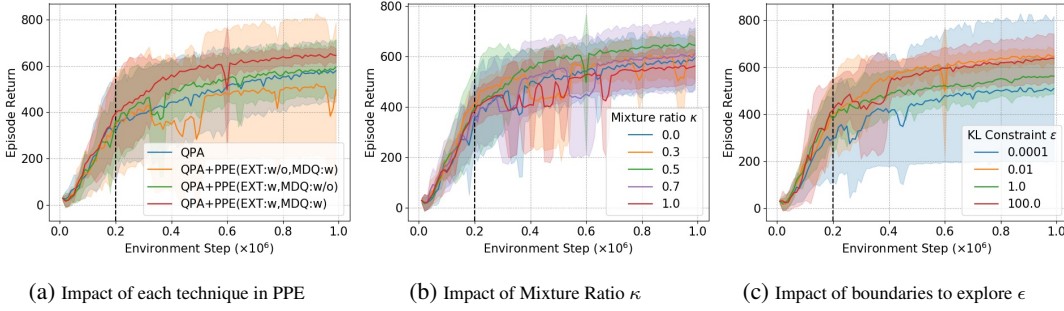

(a) Impact of each technique in PPE   (b) Impact of Mixture Ratio $\kappa$   (c) Impact of boundaries to explore $\epsilon$

Figure 4: Various ablation studies on the Cheetah-run task. The dashed black line indicates the final feedback collection step. Shaded areas indicate standard deviation; the solid line shows the mean.

Due to space limitations in the main text, we present additional ablation studies conducted on the Walker-run and Cheetah-run environments in this section. The results of these experiments are shown in Figures 3 and 4, respectively.

The results from these additional environments corroborate the findings presented in Section 4.2. As shown in Figures 3(a) and 4(a), the full PPE method (MDQ+EXT) consistently outperforms variants using only partial components, as well as the baseline without PPE.

Furthermore, the hyperparameter sensitivity analyses reinforce our choices. Figures 3(b), 4(b), 3(c), and 4(c) confirm that setting the mixture ratio $\kappa = 0.5$ and the KL constraint $\epsilon = 0.01$ yields robustly superior performance. Although other parameter values occasionally achieve comparable results, this combination proves to be the most stable and effective across different tasks.

As discussed in Section 4.2, these hyperparameter choices are well-grounded. A moderate value for $\kappa$ is crucial for the reward model's effectiveness, as it strikes a critical balance: it prevents the reward model from being undertrained on in-distribution trajectories, while also avoiding overfitting to out-of-distribution data. Similarly, a moderate $\epsilon$ ensures meaningful exploration (EXT) without sacrificing theoretical performance guarantees. Finally, it is worth noting that we do not perform an ablation study on the entropy coefficient of the underlying SAC algorithm, as our implementation utilizes its automated tuning version [46].

### E.7 Coverage Visualization

We collected 100 feedback instances during the learning process of the Walker_walk task using the QPA and QPA+PPE methods. The state and action spaces of these (s, a) pairs were clustered into 10 and 20 groups, respectively, using KMeans. We then used heatmaps to illustrate how the coverage of the preference buffer changes as feedback increases.

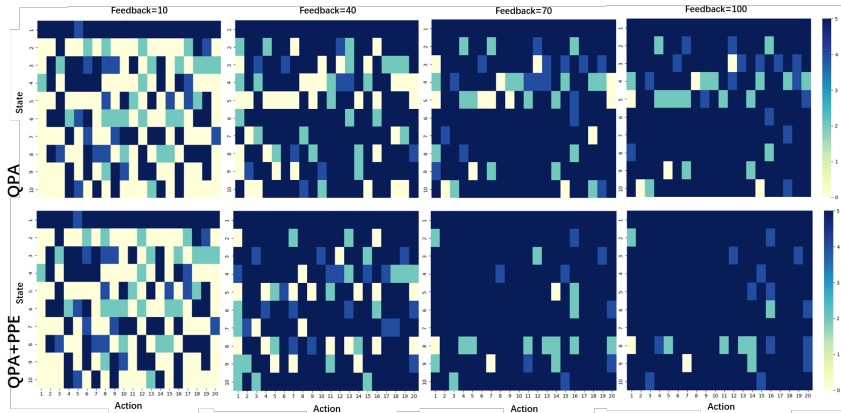

Figure 5: Distribution of actions in different discrete states after clustering. The horizontal axis represents the 20 clustered actions, and the vertical axis represents the 10 clustered states. The first and second rows show the changes in coverage of the preference buffer during training for the QPA and QPA+PPE methods, respectively.

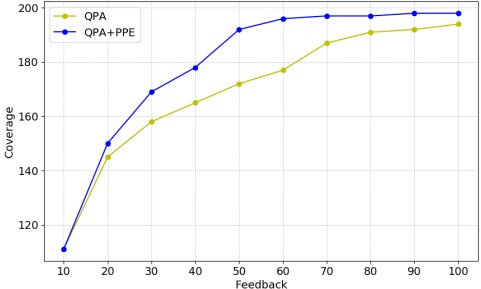

Figure 6: Changes in coverage of the 10×20 clustered (s, a) space for the preference buffer corresponding to the QPA and QPA+PPE methods under the same seed.

Figures 5 and 6 demonstrate that as the number of queries increases, the use of PQPA+PPE clearly enhances coverage compared to QPA.

## E.8  Human Feedback Experiments

**How to Obtain Genuine Human Preferences Online**  We achieve authentic interaction with humans in the process of obtaining human preferences through the code above. This involves presenting two sets of behavior segment videos to humans and requesting preference labels from them. The specific interaction interface is shown in Figure 7.

**Collecting Human Feedback**

```python
import imageio as iio

def get_label(self, sa_t_1, sa_t_2, physics_seg1, physics_seg2):

    frame_height, frame_width, channels = physics_seg1[0,0].shape

    # Create a video writer
    output_width = frame_width * 2  # The merged width is twice the original.
    output_height = frame_height
    fps = 30  # Set the frame rate.

    # Save video
    human_labels = np.zeros(sa_t_1.shape[0])
    for seg_index in range(physics_seg1.shape[0]):
        # render the pairs of segments and save the video
        # Create a video writer using imageio
        with iio.get_writer(f'output.mp4', fps=fps) as writer:
            # Iterate over all frames.
            for frame0, frame1 in zip(physics_seg1[seg_index], physics_seg2[seg_index]):
                # Horizontally merge frames
                combined_frame = np.hstack((frame0, frame1))
                # Write to the video file
                writer.append_data(combined_frame)
        labeling = True
        # provide labeling instruction and query human for preferences
        while(labeling):
            print("\n")
            print("--------------------------------------------------")
            print("Feedback number:", seg_index)
            # preference:
            # 0: segment 0 is better
            # 1: segment 1 is better
            while True:
                # check if it is 0/1/number type preference
                try:
                    rational_label = input("Preference: 0 or 1 or other number")
                    rational_label = int(rational_label)
                    break
                except:
                    print("Wrong label type. Please enter 0/1/other number.")
            print("--------------------------------------------------")
            human_labels[seg_index] = rational_label
            labeling = False
    #remove the hard-to-judge pairs of segments
    cancel = np.where((human_labels != 0) & (human_labels != 1))[0]
    human_labels = np.delete(human_labels, cancel, axis=0)
    sa_t_1 = np.delete(sa_t_1, cancel, axis=0)
    sa_t_2 = np.delete(sa_t_2, cancel, axis=0)
    print("valid query number:", len(human_labels))
    return sa_t_1, sa_t_2, human_labels.reshape(-1,1)
```

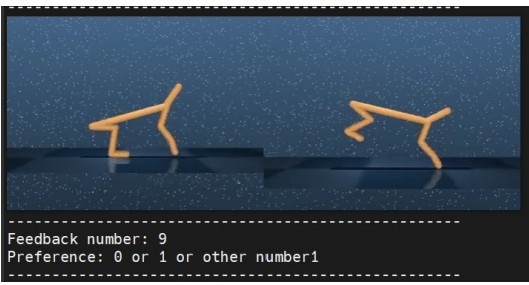

Figure 7: Through this interface, humans can provide preference labels for the agent's behavior.

**Comparative Performance Analysis of QPA and PPE in Human Preference Experiments**  To empirically validate our algorithm's effectiveness, we conducted human preference experiments with five volunteer annotators. Each volunteer was asked to provide preference labels based on videos of agents performing the cheetah-run task. Specifically, each annotator completed 100 preference annotations, organized into 10-label batches. After every 20,000 training steps, a new batch of labels was collected. Notably, this experimental setting is fully consistent with the preference labeling setup described in the paper for the cheetah-run task, as detailed in Table 10

As illustrated in Figure 8, which presents mean performance values (solid lines) with standard deviation ranges (shaded areas), our analysis reveals that PPE demonstrates superior performance compared to QPA. To facilitate comparative analysis, we have included demonstration videos of the final trained agents in the supplementary materials, showcasing the practical behavioral differences between PPE-trained and QPA-trained agents.

Our investigation further uncovered a critical distinction in annotation: the QPA-provided queries exhibited substantial similarity in trajectory samples (Figure 9(a)). This phenomenon may be attributed to QPA reducing the size of $\mathcal{D}^{cp}$, which consequently diminishes trajectory diversity in $\mathcal{D}^{cp}$ - a configuration that increases cognitive load and decision uncertainty for human annotators.

In contrast, PPE's active exploration mechanism systematically expands the policy-proximal sampling space, as evidenced in Figure 9(b). This strategic diversity enhancement in query selection (1) improves sample representativeness, and (2) reduces the cognitive burden on human annotators by presenting more distinguishable trajectory pairs. We identify this fundamental difference in query as a primary contributor to PPE's performance advantage in this human preference experiments.

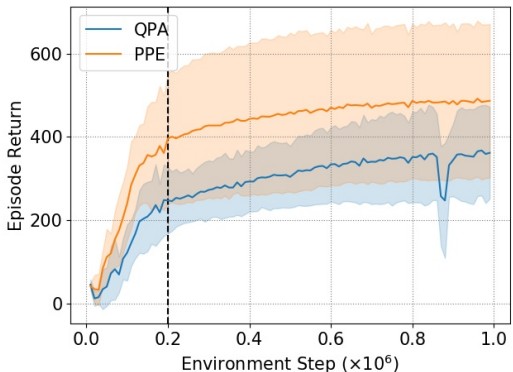

Figure 8: Performance comparison between PPE and QPA on the Cheetah-run task using real human preference feedback.

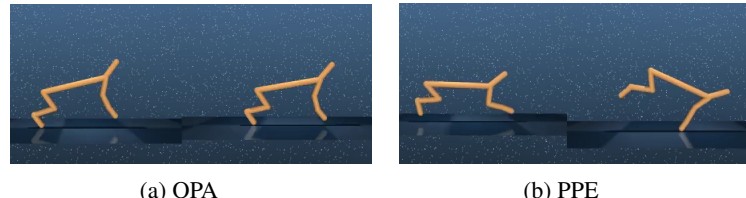

(a) QPA                                    (b) PPE

Figure 9: Comparative visualization of trajectory pair selection during human preference queries. **(a)** QPA's homogeneous sampling results in human preference queries. **(b)** PPE's active exploration generates diverse, distinguishable samples.

# F  About OOD Detection

## F.1  Why Morse Network?

**Theoretical Analysis**  Our core investigation focuses on whether increasing the preference buffer coverage can optimize the performance of online PbRL.

This is supported by the motivation example in Section 3.1, the coverage visualization in Appendix E.7, and extensive experiments throughout the paper.

The Morse network in our work is only used to provide gradients to $\pi_E$ for exploring OOD data.

Compared to ensemble-based OOD detection mechanisms like RND [47], the Morse network offers several advantages:

(1). The Morse network requires only a single neural network, whereas RND requires multiple networks for OOD detection, thus reducing computational overhead.

(2). The output of the Morse network is within the range [0,1]. In contrast, the output of RND is derived from the difference between the neural network's output and a random network, and this output is not standardized. Specifically, the form of the Morse network used in this paper is:

$$\min_\phi \frac{1}{N} \sum_{s,a \sim \mathcal{D}^p} \left[ \frac{\lambda^2}{2} ||f_\phi(s,a) - a||^2 + \frac{1}{M} \sum_{a_u \sim \mathrm{Uniform}(\mathcal{A})} \exp^{-\frac{\lambda^2}{2}||f_\phi(s,a_u) - a_u||^2} \right] \quad (24)$$

If we were to use the RND form, it would be:

$$\min_\phi \frac{1}{N} \sum_{s,a \sim \mathcal{D}^p} \left[ ||f_\phi(s,a) - f_{\mathrm{random}}(s,a)||^2 \right] \quad (25)$$

By setting the RND's random network to $f_{\mathrm{random}}(s,a) = a$, it matches the left part of the optimization objective of the Morse network used in our paper.

Therefore, the Morse network has the additional objective of *moving away from out-of-distribution (s,a)* and requires fewer neural networks compared to RND.

**Experimental Analysis**   To strengthen the paper by including experimental results using other OOD detection mechanisms to demonstrate that PPE can be applied to other OOD detection methods, we have added a set of experimental results using RND, as shown below.

Table 8: Comparison of PPE algorithm performance using different OOD detection methods.

| Task | Morse Network | RND |
|------|---------------|-----|
| drawer-open | **79.61 $\pm$ 44.55** | 79.52 $\pm$ 44.53 |
| Sweep-into | **96.47 $\pm$ 8.47** | 77.17 $\pm$ 20.69 |
| Hammer | **96.27 $\pm$ 5.19** | 77.41 $\pm$ 16.71 |

### F.2   Computational Cost

**Discussion on $f_\phi$**   Firstly, In our study, we utilized a neural network with a 3x256 architecture to learn the function $f_\phi$ required for the Morse network, as described in Eq.(24).

Secondly, we do not rely on the specific outputs of the Morse network to determine whether data is OOD. Instead, we only utilize the gradient $\nabla_a M_\phi(s,a)$ and use it as a basis for sampling data in the '*Mixture Distribution Query*'. These applications do not demand high precision in the Morse network's outputs; they only require a relative distinction in magnitude between in-distribution and out-of-distribution data.

Lastly, Given that our dataset is not very large, especially when QPA is used as the backbone with a dataset size of only '10 $\times$ episode_length', which does not impose significant stress on the neural network.

Considering computational costs, we only train the Morse network for an additional 200 iterations after completing after per query. It is noteworthy that in many tasks, QPA and SURF involve training the reward model thousands of times after per query. Therefore, our use of the Morse network effectively meets our needs without incurring substantial additional computational overhead.

**Experiments Results of Computational Cost**   We averaged the time required for QPA and QPA+PPE to train the reward model (and the Morse network) after five query phases on the walker_walk task, all conducted on the same machine.

Table 9: Average time comparison between QPA and QPA+PPE.

| Method | Average Time (seconds) |
| --- | --- |
| QPA | 45.29 |
| QPA+PPE | 50.42 |

While PPE does introduce additional computational overhead, training the Morse network, like the reward model, is only necessary after each query. The total number of queries varies by task. For instance, in the 'walker_walk' task, we followed the QPA setup, requiring a total of 100 preference feedbacks, with each query obtaining 10 preference feedbacks. Therefore, the overall training process does not significantly increase computational cost.

# G   More Details About The Motivating Example In Section 3.1

## G.1   Meaning of Region $1 - 9$

In Section 3.1, "region 1-9" refers to square regions depicted in Figure 1(a), with the lower-left corner as the origin. The grid is labeled from 0 to 9 on both the horizontal and vertical axes, increasing from left to right and from bottom to top, respectively. For example, "region 3" denotes a grid area bounded by the segments from 0 to 3 on both axes.

## G.2   The Evaluation Region for Used in Figure 1(d)

The evaluation region used is the same as the training region. This figure is intended to explore how varying the amount of preference feedback affects the performance of the reward model when both the evaluation and training regions are fixed.

# H   Implementation Details

## H.1   Fundamental Process of PbRL

An overview of the components in a typical PbRL setup can be provided as below:

    a). Data collection

    b). Data selection and preference labeling

    c). Learning the reward model using preference labels $(\tau^0, \tau^1, y_p)$

    d). Optimizing $\pi_T$ with the learned reward model via reinforcement learning methods

## H.2   About Buffers

### H.2.1   Functions of Various Buffers

- $\mathcal{D}^{cp}$ stores potential segments $\tau$ that might be selected during the "data selection and preference labeling" phase. Specifically, when selecting $(\tau^0, \tau^1)$ for preference labeling, these segments are drawn from $\mathcal{D}^{cp}$.

- $\mathcal{D}$ is the replay buffer, a fundamental concept in reinforcement learning, storing $(s_t, a_t, \hat{r}_t, s_{t+1})$ instead of the ground truth $r_t$ . It is used during the policy optimization phase with the learned reward model.

- $\mathcal{D}^p$ stores preference feedbacks $(\tau^0, \tau^1, y_p)$ for learning the reward model.

- $\mathcal{D}^m$ stores an additional one-dimensional data $M_\phi(s, a)$ for each $(s, a)$ in $\mathcal{D}^{cp}$, as shown in Eq. 5. It is used to compute to assess the OOD degree of $\tau$.

### H.2.2 Memory Usage

- $\mathcal{D}$ is essential for all off-policy reinforcement learning algorithms as a replay buffer.

- $\mathcal{D}^{cp}$ and $\mathcal{D}^p$ are necessary for existing online PbRL methods.

- $\mathcal{D}^m$ only requires storing an additional one-dimensional value $M_\phi(s, a)$ for each $(s, a)$ in $\mathcal{D}^{cp}$, which is a minor addition performed in Algorithm 2, line 4

Therefore, PPE does not require significantly more memory compared to previous online PbRL methods.

### H.3 Origin of the Code for Baseline Algorithms

To ensure fairness in our experiments, we used the original source code provided by the authors of each baseline algorithm. Specifically, the sources are as follows:

- PEBBLE, SURF:
  `https://openreview.net/attachment?id=TfhfZLQ2EJO&name=supplementary_material`
- RUNE:`https://github.com/rll-research/rune`
- QPA:`https://github.com/huxiao09/QPA`
- B-Pref: `https://github.com/rll-research/BPref`

The only modification we made was to unify the logging format during training. We changed QPA's logging from using wandb to the storage format used by the B-Pref framework, which is also used by PEBBLE, SURF, and RUNE.

### H.4 Human Involvement

In stage **b**, algorithms typically select $(\tau^0, \tau^1)$ pairs, which are then submitted for human preference labeling. In most PbRL implementations, scripts are typically used to simulate human preference labeling. Our paper follows the same setup.

The Mixture Distribution Query is used only in stage b to select , as shown in Algorithm 1. These selected pairs are then submitted for human preference labeling (Algorithm 2, line 8). This is the only stage that requires human involvement.

This process is consistent with what is described in PEBBLE (Algorithm 2, line 11), QPA [5] (Algorithm 1, line 6), and RUNE (Algorithm 1, line 9).

### H.5 How Were Preferences Elicited?

We used the same approach as PEBBLE, SURF, RUNE, and QPA, utilizing the B-pref framework [16] to script access to the ground truth reward, thereby simulating human preference labels.

### H.6 Reproducibility and Compute Resource Details

To ensure full reproducibility of our experiments, we provide comprehensive details on the computing environment used throughout our work. These details address typical concerns regarding the type of compute workers, memory, storage, and execution times, as outlined below:

- **Hardware Specifications:** We conducted all experiments on a dedicated server featuring an Intel Xeon processor with 64 cores. The server is equipped with 512GB of RAM and 10TB of storage, ensuring that the dataset loading and large-scale computations can be handled efficiently.

- **GPU Configuration:** The experimental environment includes eight NVIDIA 1080ti GPUs, each with 11GB of VRAM. Each training task is assigned an individual GPU, although multiple tasks may share a single GPU simultaneously to optimize resource utilization. This configuration is essential for achieving the reported experimental throughput.

- **Compute Time:** On average, each task requires approximately 8 hours of training time. This metric provides a clear indication of the computational budget needed for individual experimental runs. Notably, besides the standard training procedures detailed in our paper, no additional compute was required, as preliminary or failed experiments did not necessitate extra resources beyond what is reported.

By providing these explicit details, we aim to facilitate a clear understanding of the computational resources required to reproduce our experiments and assess the scalability and feasibility of the approach on similar hardware setups.

### H.7 Crowdsourcing and Human Subject Research Details

**Overview** In our study, we conducted experiments involving human subjects to collect preference labels for agent behaviors. Volunteers were recruited to participate in these experiments, and appropriate compensation was provided in accordance with the NeurIPS Code of Ethics, ensuring that all participants received at least the minimum wage of the country in which the data was collected.

**Instructions and Interaction Interface** Participants were presented with pairs of behavior segment videos and asked to indicate their preference between the two. The full text of the instructions provided to participants is included in the supplementary material. The interaction was facilitated through a custom interface, as illustrated in Figure 7. This interface displayed two videos side by side, and participants were prompted to select the segment they felt better matched the task description of "controlling the robot to run like an animal," or to indicate if neither segment was clearly preferable.

Figure 7 shows the actual interface used for collecting human feedback. Participants were instructed as follows:

> You will be shown two short video segments of agent behavior. Please watch both videos and indicate which one you prefer by entering '0' if you prefer the left video, '1' if you prefer the right video, or another number if you cannot decide. If you cannot make a clear choice, the pair will be excluded from the analysis.

**Compensation** All participants were compensated for their time and effort. The compensation rate was set to ensure compliance with ethical standards and local minimum wage regulations.

**Ethical Considerations** This study was conducted in accordance with the NeurIPS Code of Ethics. All participants provided informed consent prior to participation, and their data was anonymized to protect privacy.

### H.8 Parameter for Tasks

Table 10: The hyperparameters of tasks

| Hyper-parameter | Total feedback | Frequency of feedback | Queries number per session | Training Steps |
|---|---|---|---|---|
| Walker-walk | 1e2 | 2e4 | 1e1 | 5e5 |
| Walker-run | 1e2 | 2e4 | 1e1 | 1e6 |
| Cheetah-run | 1e2 | 2e4 | 1e1 | 1e6 |
| Quadruped-walk | 1e3 | 3e4 | 1e2 | 1e6 |
| Quadruped-run | 1e3 | 3e4 | 1e2 | 1e6 |
| Humanoid-stand | 1e4 | 5e3 | 5e1 | 2e6 |
| Drawer-open | 4e3 | 5e3 | 2e1 | 1e6 |
| Sweep-into | 1e4 | 5e3 | 5e1 | 2e6 |
| Hammer | 1e4 | 5e3 | 5e1 | 2e6 |
| Door-open | 3e3 | 5e3 | 3e1 | 1e6 |
| Door-unlock | 3e3 | 5e3 | 3e1 | 1e6 |
| Button-press | 2e3 | 5e3 | 25 | 1e6 |
| Window-open | 4e2 | 5e3 | 1e1 | 5e5 |

Determining the number of feedback instances for each task, the interval between queries, and the quantity of feedback per query can be quite challenging. We have summarized the experimental settings from the QPA [21] and SURF [14] papers in Table 10. The experiments in our paper strictly adhere to the settings outlined in this table.

## H.9 Parameter for Algorithms

Our method does not introduce many additional parameters, as shown in Table 11. In this work, $\epsilon$ represents the KL divergence constraint between the behavior policy and the target policy in Eq.(3), which determines the exploration boundary in our approach. The parameter $\lambda$ controls the sensitivity of the Morse Neural Network. Lastly, $\kappa$, mentioned in Algorithm 2, is the mixture ratio that controls the proportion of samples drawn from each distribution. Additionally, we followed the parameter

Table 11: The hyperparameters of PPE

| Hyper-parameter | Value | Hyper-parameter | Value |
| --- | --- | --- | --- |
| KL constraint $\epsilon$ | 1e-2 | Parameter for OOD detection $\lambda$ | 5 |
| Mixture ratio $\kappa$ | 0.5 | | |

settings from the baseline papers [21, 12, 14, 15, 13]. The specific parameter configurations are detailed in Tables 12 to 15.

Table 12: The hyperparameters of SAC

| Hyper-parameter | Value | Hyper-parameter | Value |
| --- | --- | --- | --- |
| Discount | 0.99 | Init temperature | 0.1 |
| Alpha learning rate | 1e-4 | Batch size | 1024 |
| Critic target update freq | 2 | Critic EMA | 5e-3 |
| Critic learning rate | 5e-4 (Walker_walk, Cheetah_run, Walker_run) 1e-4 (Other tasks) | Actor learning rate | 5e-4 (Walker_walk, Cheetah_run, Walker_run) 1e-4 (Other tasks) |
| Critic hidden dim | 1024 | Actor hidden dim | 1024 |
| Critic hidden layer | 2 | Actor hidden layer | 2 |
| Critic activation function | ReLU | Actor activation function | ReLU |
| Optimizer | Adam | | |

Table 13: The hyperparameters of QPA

| Hyper-parameter | Value | Hyper-parameter | Value |
| --- | --- | --- | --- |
| Size of policy-aligned buffer $N$ | 10 | Data augmentation ratio $\tau$ | 20 |
| Hybrid experience replay sample ratio $\omega$ | 0.5 | Min/Max length of subsampled snippets | [35, 45] |

Table 14: The hyperparameters of SURF

| Hyper-parameter | Value | Hyper-parameter | Value |
| --- | --- | --- | --- |
| Unlabeled batch ratio | 4 | Threshold | 0.99 |
| Loss weight | 1 | Min/Max length of cropped segment | [45, 55] |
| Segment length before cropping | 60 | | |

Table 15: The hyperparameters of PEBBLE

| Hyper-parameter | Value | Hyper-parameter | Value |
| --- | --- | --- | --- |
| Length of segment | 50 | Unsupervised pre-training steps | 9000 |
| Size of query selection buffer | 100 | | |

# NeurIPS Paper Checklist

1. **Claims**

   Question: Do the main claims made in the abstract and introduction accurately reflect the paper's contributions and scope?

   Answer: [Yes]

   Justification: The main claims in the abstract and introduction accurately reflect the paper's contributions and scope, clearly highlighting the challenge of limited preference buffer coverage in PbRL and presenting PPE as a principled solution with demonstrated empirical benefits. The stated contributions are consistent with the methods and results described.

   Guidelines:
   - The answer NA means that the abstract and introduction do not include the claims made in the paper.
   - The abstract and/or introduction should clearly state the claims made, including the contributions made in the paper and important assumptions and limitations. A No or NA answer to this question will not be perceived well by the reviewers.
   - The claims made should match theoretical and experimental results, and reflect how much the results can be expected to generalize to other settings.
   - It is fine to include aspirational goals as motivation as long as it is clear that these goals are not attained by the paper.

2. **Limitations**

   Question: Does the paper discuss the limitations of the work performed by the authors?

   Answer: [Yes]

   Justification: The paper discusses limitations in the "Conclusion and Discussion" section. Specifically, authors acknowledge that current query method does not account for the variations in information between different pairs of agent behaviors. They also mention that their approach could be further improved by considering factors such as data similarity and clustering traits to refine and optimize the query method. This reflects an awareness of the limitations in feedback efficiency and the potential for further enhancement of this method.

   Guidelines:
   - The answer NA means that the paper has no limitation while the answer No means that the paper has limitations, but those are not discussed in the paper.
   - The authors are encouraged to create a separate "Limitations" section in their paper.
   - The paper should point out any strong assumptions and how robust the results are to violations of these assumptions (e.g., independence assumptions, noiseless settings, model well-specification, asymptotic approximations only holding locally). The authors should reflect on how these assumptions might be violated in practice and what the implications would be.
   - The authors should reflect on the scope of the claims made, e.g., if the approach was only tested on a few datasets or with a few runs. In general, empirical results often depend on implicit assumptions, which should be articulated.
   - The authors should reflect on the factors that influence the performance of the approach. For example, a facial recognition algorithm may perform poorly when image resolution is low or images are taken in low lighting. Or a speech-to-text system might not be used reliably to provide closed captions for online lectures because it fails to handle technical jargon.
   - The authors should discuss the computational efficiency of the proposed algorithms and how they scale with dataset size.
   - If applicable, the authors should discuss possible limitations of their approach to address problems of privacy and fairness.
   - While the authors might fear that complete honesty about limitations might be used by reviewers as grounds for rejection, a worse outcome might be that reviewers discover limitations that aren't acknowledged in the paper. The authors should use their best

judgment and recognize that individual actions in favor of transparency play an important role in developing norms that preserve the integrity of the community. Reviewers will be specifically instructed to not penalize honesty concerning limitations.

3. **Theory assumptions and proofs**

Question: For each theoretical result, does the paper provide the full set of assumptions and a complete (and correct) proof?

Answer: [Yes]

Justification: The paper provides detailed theoretical derivations and proofs for its main results. Specifically, in the Appendix D "Proof of Proposition 3.1", the paper clearly states the optimization problem, the assumptions (such as the use of Gaussian distributions and the tightening of the KL divergence constraint), and provides a step-by-step derivation leading to a closed-form solution. The derivation includes the construction of the Lagrangian, application of the KKT conditions, and a Taylor expansion to justify the approximation used. All key formulas are numbered and cross-referenced. The assumptions (e.g., setting $\Sigma = \Sigma_T$ for tractability and real-time requirements) are explicitly stated and discussed in the context of the proof. Therefore, the paper meets the requirements for providing a full set of assumptions and a complete and correct proof for its theoretical results.

Guidelines:

- The answer NA means that the paper does not include theoretical results.
- All the theorems, formulas, and proofs in the paper should be numbered and cross-referenced.
- All assumptions should be clearly stated or referenced in the statement of any theorems.
- The proofs can either appear in the main paper or the supplemental material, but if they appear in the supplemental material, the authors are encouraged to provide a short proof sketch to provide intuition.
- Inversely, any informal proof provided in the core of the paper should be complemented by formal proofs provided in appendix or supplemental material.
- Theorems and Lemmas that the proof relies upon should be properly referenced.

4. **Experimental result reproducibility**

Question: Does the paper fully disclose all the information needed to reproduce the main experimental results of the paper to the extent that it affects the main claims and/or conclusions of the paper (regardless of whether the code and data are provided or not)?

Answer: [Yes]

Justification: The paper provides all necessary information to enable reproducibility of the main experimental results. Specifically, Algorithm2 presents the detailed algorithmic implementation, AppendixH describes the experimental setup and implementation details, and Appendix E.8 explains how real human preference feedback is incorporated. These sections together ensure that readers have access to the full methodology, parameter settings, and procedures required to replicate the experiments and verify the main claims and conclusions of the paper.

Guidelines:

- The answer NA means that the paper does not include experiments.
- If the paper includes experiments, a No answer to this question will not be perceived well by the reviewers: Making the paper reproducible is important, regardless of whether the code and data are provided or not.
- If the contribution is a dataset and/or model, the authors should describe the steps taken to make their results reproducible or verifiable.
- Depending on the contribution, reproducibility can be accomplished in various ways. For example, if the contribution is a novel architecture, describing the architecture fully might suffice, or if the contribution is a specific model and empirical evaluation, it may be necessary to either make it possible for others to replicate the model with the same dataset, or provide access to the model. In general. releasing code and data is often one good way to accomplish this, but reproducibility can also be provided via detailed instructions for how to replicate the results, access to a hosted model (e.g., in the case

of a large language model), releasing of a model checkpoint, or other means that are appropriate to the research performed.

- While NeurIPS does not require releasing code, the conference does require all submissions to provide some reasonable avenue for reproducibility, which may depend on the nature of the contribution. For example

    (a) If the contribution is primarily a new algorithm, the paper should make it clear how to reproduce that algorithm.

    (b) If the contribution is primarily a new model architecture, the paper should describe the architecture clearly and fully.

    (c) If the contribution is a new model (e.g., a large language model), then there should either be a way to access this model for reproducing the results or a way to reproduce the model (e.g., with an open-source dataset or instructions for how to construct the dataset).

    (d) We recognize that reproducibility may be tricky in some cases, in which case authors are welcome to describe the particular way they provide for reproducibility. In the case of closed-source models, it may be that access to the model is limited in some way (e.g., to registered users), but it should be possible for other researchers to have some path to reproducing or verifying the results.

5. **Open access to data and code**

    Question: Does the paper provide open access to the data and code, with sufficient instructions to faithfully reproduce the main experimental results, as described in supplemental material?

    Answer: [Yes]

    Justification: The code will be made publicly available after the paper is accepted, ensuring open access for reproducibility. Additionally, the supplemental material includes videos of agent behaviors learned from real human preference labels provided by volunteers, which serve as qualitative evidence of the method's effectiveness.

    Guidelines:

    - The answer NA means that paper does not include experiments requiring code.
    - Please see the NeurIPS code and data submission guidelines (`https://nips.cc/public/guides/CodeSubmissionPolicy`) for more details.
    - While we encourage the release of code and data, we understand that this might not be possible, so "No" is an acceptable answer. Papers cannot be rejected simply for not including code, unless this is central to the contribution (e.g., for a new open-source benchmark).
    - The instructions should contain the exact command and environment needed to run to reproduce the results. See the NeurIPS code and data submission guidelines (`https://nips.cc/public/guides/CodeSubmissionPolicy`) for more details.
    - The authors should provide instructions on data access and preparation, including how to access the raw data, preprocessed data, intermediate data, and generated data, etc.
    - The authors should provide scripts to reproduce all experimental results for the new proposed method and baselines. If only a subset of experiments are reproducible, they should state which ones are omitted from the script and why.
    - At submission time, to preserve anonymity, the authors should release anonymized versions (if applicable).
    - Providing as much information as possible in supplemental material (appended to the paper) is recommended, but including URLs to data and code is permitted.

6. **Experimental setting/details**

    Question: Does the paper specify all the training and test details (e.g., data splits, hyperparameters, how they were chosen, type of optimizer, etc.) necessary to understand the results?

    Answer: [Yes]

Justification: The paper provides a clear description of the experimental setup in the main text (Section 4), including the benchmarks used (MetaWorld and DMControl), the tasks evaluated, the baselines for comparison, the use of five random seeds, and the evaluation metrics (mean and standard deviation of the final 10 evaluation episodes). For further transparency and reproducibility, the paper refers readers to Appendix H, which contains comprehensive details on experimental settings, such as hyperparameters, training procedures, and implementation specifics. This ensures that all necessary information to understand and interpret the experimental results is available.

Guidelines:

- The answer NA means that the paper does not include experiments.
- The experimental setting should be presented in the core of the paper to a level of detail that is necessary to appreciate the results and make sense of them.
- The full details can be provided either with the code, in appendix, or as supplemental material.

7. **Experiment statistical significance**

Question: Does the paper report error bars suitably and correctly defined or other appropriate information about the statistical significance of the experiments?

Answer: [Yes]

Justification: The paper reports the mean and standard deviation of the experimental results across five random seeds. Additionally, Appendix E presents the p-values from Welch's T-test comparing the performance of the PPE algorithm with other baselines, offering a formal assessment of statistical significance. These practices ensure that the reported results are statistically sound and that the main claims are supported by appropriate statistical analysis.

Guidelines:

- The answer NA means that the paper does not include experiments.
- The authors should answer "Yes" if the results are accompanied by error bars, confidence intervals, or statistical significance tests, at least for the experiments that support the main claims of the paper.
- The factors of variability that the error bars are capturing should be clearly stated (for example, train/test split, initialization, random drawing of some parameter, or overall run with given experimental conditions).
- The method for calculating the error bars should be explained (closed form formula, call to a library function, bootstrap, etc.)
- The assumptions made should be given (e.g., Normally distributed errors).
- It should be clear whether the error bar is the standard deviation or the standard error of the mean.
- It is OK to report 1-sigma error bars, but one should state it. The authors should preferably report a 2-sigma error bar than state that they have a 96% CI, if the hypothesis of Normality of errors is not verified.
- For asymmetric distributions, the authors should be careful not to show in tables or figures symmetric error bars that would yield results that are out of range (e.g. negative error rates).
- If error bars are reported in tables or plots, The authors should explain in the text how they were calculated and reference the corresponding figures or tables in the text.

8. **Experiments compute resources**

Question: For each experiment, does the paper provide sufficient information on the computer resources (type of compute workers, memory, time of execution) needed to reproduce the experiments?

Answer: [Yes]

Justification: Due to space limitations in the main text, detailed information regarding the comprehensive descriptions of the hardware environment will be provided in the appendix. This ensures that all necessary details are available to reproduce the experiments fully, while keeping the main paper concise.

Guidelines:

- The answer NA means that the paper does not include experiments.
- The paper should indicate the type of compute workers CPU or GPU, internal cluster, or cloud provider, including relevant memory and storage.
- The paper should provide the amount of compute required for each of the individual experimental runs as well as estimate the total compute.
- The paper should disclose whether the full research project required more compute than the experiments reported in the paper (e.g., preliminary or failed experiments that didn't make it into the paper).

9. **Code of ethics**

Question: Does the research conducted in the paper conform, in every respect, with the NeurIPS Code of Ethics https://neurips.cc/public/EthicsGuidelines?

Answer: [Yes]

Justification: The research presented in this paper adheres fully to the NeurIPS Code of Ethics. The work involves algorithmic development and simulation experiments on standard continuous control benchmarks. There are no foreseeable risks of harm, privacy violations, or misuse of the technology, and does not engage in any discriminatory or unethical practices. All data and code used comply with relevant ethical standards.

Guidelines:

- The answer NA means that the authors have not reviewed the NeurIPS Code of Ethics.
- If the authors answer No, they should explain the special circumstances that require a deviation from the Code of Ethics.
- The authors should make sure to preserve anonymity (e.g., if there is a special consideration due to laws or regulations in their jurisdiction).

10. **Broader impacts**

Question: Does the paper discuss both potential positive societal impacts and negative societal impacts of the work performed?

Answer: [Yes]

Justification: This paper discusses the potential positive societal impacts of improving the efficiency and reliability of preference-based reinforcement learning (PbRL). Such advancements can enable the development of more effective and human-aligned AI systems across various domains, including robotics, personalized assistance, and autonomous systems. By reducing the human effort required to design reward functions and enhancing policy learning from human preferences, this work contributes to making AI systems more adaptable and user-friendly.

At the same time, we acknowledge potential negative societal impacts. For instance, if preference-based RL systems are deployed without proper oversight, they may learn biased or unsafe behaviors due to flawed or insufficient human feedback. Moreover, misuse of these systems could result in unintended consequences, especially in safety-critical applications. We recognize the importance of managing preference buffer coverage and ensuring reliable reward modeling as essential measures to mitigate such risks.

Guidelines:

- The answer NA means that there is no societal impact of the work performed.
- If the authors answer NA or No, they should explain why their work has no societal impact or why the paper does not address societal impact.
- Examples of negative societal impacts include potential malicious or unintended uses (e.g., disinformation, generating fake profiles, surveillance), fairness considerations (e.g., deployment of technologies that could make decisions that unfairly impact specific groups), privacy considerations, and security considerations.
- The conference expects that many papers will be foundational research and not tied to particular applications, let alone deployments. However, if there is a direct path to any negative applications, the authors should point it out. For example, it is legitimate to point out that an improvement in the quality of generative models could be used to

generate deepfakes for disinformation. On the other hand, it is not needed to point out that a generic algorithm for optimizing neural networks could enable people to train models that generate Deepfakes faster.

- The authors should consider possible harms that could arise when the technology is being used as intended and functioning correctly, harms that could arise when the technology is being used as intended but gives incorrect results, and harms following from (intentional or unintentional) misuse of the technology.
- If there are negative societal impacts, the authors could also discuss possible mitigation strategies (e.g., gated release of models, providing defenses in addition to attacks, mechanisms for monitoring misuse, mechanisms to monitor how a system learns from feedback over time, improving the efficiency and accessibility of ML).

11. **Safeguards**

Question: Does the paper describe safeguards that have been put in place for responsible release of data or models that have a high risk for misuse (e.g., pretrained language models, image generators, or scraped datasets)?

Answer: [NA]

Justification: The paper does not involve the release of models or datasets that pose a high risk of misuse. The work focuses on reinforcement learning methods and experiments using standard benchmark environments without releasing pretrained models or scraped datasets. Therefore, no specific safeguards for responsible release are necessary.

Guidelines:

- The answer NA means that the paper poses no such risks.
- Released models that have a high risk for misuse or dual-use should be released with necessary safeguards to allow for controlled use of the model, for example by requiring that users adhere to usage guidelines or restrictions to access the model or implementing safety filters.
- Datasets that have been scraped from the Internet could pose safety risks. The authors should describe how they avoided releasing unsafe images.
- We recognize that providing effective safeguards is challenging, and many papers do not require this, but we encourage authors to take this into account and make a best faith effort.

12. **Licenses for existing assets**

Question: Are the creators or original owners of assets (e.g., code, data, models), used in the paper, properly credited and are the license and terms of use explicitly mentioned and properly respected?

Answer: [Yes]

Justification: The paper uses existing assets such as benchmark environments and datasets for reinforcement learning experiments. All original creators and sources of these assets are properly cited in the paper. The versions and URLs of the assets used are clearly stated where applicable. Additionally, the licenses and terms of use for these assets have been reviewed and respected in accordance with their respective requirements.

Guidelines:

- The answer NA means that the paper does not use existing assets.
- The authors should cite the original paper that produced the code package or dataset.
- The authors should state which version of the asset is used and, if possible, include a URL.
- The name of the license (e.g., CC-BY 4.0) should be included for each asset.
- For scraped data from a particular source (e.g., website), the copyright and terms of service of that source should be provided.
- If assets are released, the license, copyright information, and terms of use in the package should be provided. For popular datasets, `paperswithcode.com/datasets` has curated licenses for some datasets. Their licensing guide can help determine the license of a dataset.

- For existing datasets that are re-packaged, both the original license and the license of the derived asset (if it has changed) should be provided.
- If this information is not available online, the authors are encouraged to reach out to the asset's creators.

13. **New assets**

Question: Are new assets introduced in the paper well documented and is the documentation provided alongside the assets?

Answer: [NA]

Justification: The paper does not introduce or release any new assets (such as datasets or models) at the time of submission. Although the authors plan to release the relevant code with detailed documentation after the paper is accepted, currently no new assets are provided. Therefore, this question is not applicable.

Guidelines:

- The answer NA means that the paper does not release new assets.
- Researchers should communicate the details of the dataset/code/model as part of their submissions via structured templates. This includes details about training, license, limitations, etc.
- The paper should discuss whether and how consent was obtained from people whose asset is used.
- At submission time, remember to anonymize your assets (if applicable). You can either create an anonymized URL or include an anonymized zip file.

14. **Crowdsourcing and research with human subjects**

Question: For crowdsourcing experiments and research with human subjects, does the paper include the full text of instructions given to participants and screenshots, if applicable, as well as details about compensation (if any)?

Answer: [Yes]

Justification: The paper reports that volunteers were recruited to provide preference labels for the human preference experiments and that appropriate compensation was given. Furthermore, the instructions provided to participants are included in the supplementary material. Therefore, the paper fulfills the requirements regarding documentation of instructions and compensation for human subjects.

Guidelines:

- The answer NA means that the paper does not involve crowdsourcing nor research with human subjects.
- Including this information in the supplemental material is fine, but if the main contribution of the paper involves human subjects, then as much detail as possible should be included in the main paper.
- According to the NeurIPS Code of Ethics, workers involved in data collection, curation, or other labor should be paid at least the minimum wage in the country of the data collector.

15. **Institutional review board (IRB) approvals or equivalent for research with human subjects**

Question: Does the paper describe potential risks incurred by study participants, whether such risks were disclosed to the subjects, and whether Institutional Review Board (IRB) approvals (or an equivalent approval/review based on the requirements of your country or institution) were obtained?

Answer: [NA]

Justification:

Although the experiment involved volunteers providing preference labels, their participation was limited to passively watching videos without any intervention or collection of sensitive personal data. The study posed minimal to no risk to participants, and no sensitive or potentially harmful procedures were involved. Therefore, formal Institutional Review Board (IRB) approval was not sought, as the research falls under minimal-risk observational studies exempt from such review according to common ethical guidelines.

Guidelines:

- The answer NA means that the paper does not involve crowdsourcing nor research with human subjects.
- Depending on the country in which research is conducted, IRB approval (or equivalent) may be required for any human subjects research. If you obtained IRB approval, you should clearly state this in the paper.
- We recognize that the procedures for this may vary significantly between institutions and locations, and we expect authors to adhere to the NeurIPS Code of Ethics and the guidelines for their institution.
- For initial submissions, do not include any information that would break anonymity (if applicable), such as the institution conducting the review.

16. **Declaration of LLM usage**

Question: Does the paper describe the usage of LLMs if it is an important, original, or non-standard component of the core methods in this research? Note that if the LLM is used only for writing, editing, or formatting purposes and does not impact the core methodology, scientific rigorousness, or originality of the research, declaration is not required.

Answer: [NA]

Justification: The algorithms and experiments in this work do not involve any large language models (LLMs), and the core method development does not rely on LLMs as important, original, or non-standard components. However, the proposed method is general and can potentially be applied to the post-training of LLMs.

Guidelines:

- The answer NA means that the core method development in this research does not involve LLMs as any important, original, or non-standard components.
- Please refer to our LLM policy (`https://neurips.cc/Conferences/2025/LLM`) for what should or should not be described.

