# OpenReview forum: "Improving Reward Models with Proximal Policy Exploration for Preference-Based Reinforcement Learning"
_NeurIPS.cc/2025/Conference — NeurIPS 2025 poster_

### Official Review · Reviewer_iwcb · 2025-06-24

**Clarity:** 2
**Significance:** 3
**Originality:** 2
**Rating:** 4
**Confidence:** 3

**Summary:**

This paper studies the problem of preference-based reinforcement learning (PbRL), in which an agent is tasked with learning a reward model based on preference relations between trajectories provided by a “teacher”, while simultaneously learning a policy that maximizes the learned reward function. The authors propose the use of two techniques for (i) expanding agent exploration in regions of the MDP which are undersampled in the agent’s buffer, and for (ii) sampling trajectories from a mixture distribution of in-distribution and OOD trajectories. To implement the first technique, the authors use Morse neural networks and introduce an approximation to a constrained optimization problem formulation. The method is evaluated on the MetaWorld and DMControl benchmarks and compared to other state-of-the-art baselines.

**Questions:**

Below, I have questions and constructive feedback to the authors:

- Section 3.2 is hard to follow and could be improved in terms of clarity and presentation. For instance, what motivates the choice of using Morse neural networks and an RBF kernel? What are the necessary assumptions for this choice to be valid? At a high level, what is the goal of Eq. 2?

- How tight is the closed-form approximate solution in Eq. 4 for Eq. 3? It would be important to provide an approximation bound in Proposition 3.1 to understand how loose this approximation is.

- In line 218, how is the dataset $D_{cp}$ collected?

- In Section 4.1, how is the preference queried? That is, who/what is ranking the trajectories?

- The paper is missing a more thorough discussion of the state-of-the-art techniques. For instance, how do PEBBLE, SURF, RUNE, and QPA differ from PPE? What is the main high-level idea behind each of these baselines?

- “... we record the mean and standard deviation of the final evaluation episodes aggregated across all random seeds”. Is this evaluated with respect to which metric? Total reward? Discounted return? The caption of Table 1 should indicate what the numbers in the table are measuring.

Regarding the claims, “As shown in Table 1, PPE consistently achieves superior performance across all benchmark tasks, indicating that PPE can more effectively select and utilize feedback under limited query budgets. This supports our claim that expanding the coverage of the preference buffer improves the reward model’s evaluation capability and leads to more reliable policy updates.”:

- Why is this setting “under a limited query budget”? This is never explained.

- The fact that PPE achieves a higher return does not necessarily imply that it is more effective in selecting and utilizing feedback, or that it is better due to the expansion of the coverage of the preference buffer. Those claims are only supported by the ablation study.

- Based on Fig. 2(a) and Fig. 2(c), it seems that the full proposed method (red curve in Fig. 2(a)) is only able to outperform QPA when the KL constraint is set to 0.01 (orange curve in Fig. 2(c)). For all other values, the performance is similar or worse than QPA (blue curve in Fig. 2(a)).
Does the recommended value of 0.01 result in better performance in other domains, or is it necessary to tune this value separately in each domain?

- Notice that some of the values in Table overlap in their [mean-std,mean+std] range, but only PPE results are in bold font, e.g., Quadruped-walk, Quadruped-run.

Other minor issues:
- Next, we trained a reward model using data from the preference buffer with a Bradley-Terry loss. -> Please add a reference.
- “ RUNELiang et al. (2022)” -> Should use \citep instead of \citet here.
- “... training process, Only w” -> incorrect capitalization after comma.

**Ethical Concerns:**

["NO or VERY MINOR ethics concerns only"]

**Final Justification:**

I thank the authors for the clarifications and detailed responses. I increased my score to 4 in light of the answers to my questions. I strongly suggest that the authors incorporate these clarifications in the main text in case of acceptance.

In my opinion, the main problem with the paper is the lack of clarity in the original submission, which generated many questions by all reviewers regarding basic definitions and algorithmic details. This is a borderline paper in my opinion. I have seen papers with worst quality being accepted and papers with better quality being rejected in the past.

**Limitations:**

Yes.

**Paper Formatting Concerns:**

I only noticed minor formatting issues (see Questions).

**Quality:**

2

**Strengths And Weaknesses:**

Strengths:
- The findings regarding the importance of having a good coverage of the state-action pairs in the agent’s buffer when learning reward models
- The proposed method, combined with the state-of-the-art method QPA, is able to outperform standard QPA (when properly tuned).

Weaknesses:
- The paper has clarity and presentation issues that make it difficult for the reader to follow it (see Questions below).
- Some algorithmic decisions need more justification and better description, as well as the experimental setting employed in Section 4 (see below).

---

> ### Author Rebuttal · Authors · 2025-07-31
>
> Dear Reviewer iwcb,
>
> We would like to extend our sincerest gratitude for your detailed and insightful review of our manuscript. Your constructive feedback is invaluable, and we believe that addressing your comments will significantly improve the quality and clarity of our paper.
>
> We are encouraged that you recognized the strengths of our work, particularly our findings on the importance of preference buffer coverage and the empirical performance improvements of our method. We have carefully considered all your suggestions and provide the following point-by-point responses to your questions and concerns.
>
> ---
>
> ### **Responses to Questions**
>
> **1. Clarity of Section 3.2 (Motivation for Morse NN, RBF Kernel, and Goal of Eq. 2)**
>
> Thank you for pointing out the need for greater clarity in this section. We agree that a clearer explanation of our methodological choices is essential.
>
> * **Motivation for Morse Neural Networks:** Our primary goal is to actively explore regions that are undersampled by the current policy but still close to it. To achieve this, we first need a mechanism to quantify whether a given state-action pair $(s, a)$ is "out-of-distribution" (OOD) with respect to the data currently in our preference buffer $\mathcal{D}^{p}$. As we discussed, Morse Neural Networks (Dherin et al., 2023)  provide a principled way to learn an unnormalized density function $M(x)$ that is explicitly designed for this purpose. The function value approaches 1 for in-distribution data (modes of the dataset) and decreases towards 0 for OOD data. This provides a direct and continuous measure of "OOD-ness," which is precisely what we need for our exploration strategy.
>
> * **Motivation for the RBF Kernel:** We use the Radial Basis Function (RBF) kernel, as shown in Eq. (1)  to shape the Morse Network. This is a standard and well-established choice in kernel-based methods (Seeger, 2004). The RBF kernel is advantageous here because its value depends on the Euclidean distance, creating a smooth, localized "bump" around in-distribution data points. This property is ideal for generating the desired density field where the density value decreases as we move away from known data, which aligns perfectly with the objective of the Morse Network.
>
> * **Goal of Equation (2):** Equation (2) defines the loss function used to train our Morse Neural Network. As we state on page 5 (lines 172-174) , this loss is derived from minimizing the KL divergence between the distribution of our preference buffer data and the learned density model. The loss function has two main components:
>     1.  The first term, $\frac{\lambda^{2}}{2}||f_{\phi}(s,a)-a||^{2}$, encourages the model to learn an identity mapping ($f_{\phi}(s,a) \approx a$) for state-action pairs $(s, a)$ that are **in-distribution** (i.e., from the preference buffer $\mathcal{D}^{p}$). This makes the Morse value $M_{\phi}(s,a)$ low, as desired for in-distribution samples.
>     2.  The second term involves sampling actions $a_{u}$ from a uniform distribution and penalizes the model if it assigns high density to these random (likely OOD) actions.
>
>     In essence, the goal of optimizing Eq. (2) is to train a model $f_{\phi}$ that can reliably distinguish between in-distribution and out-of-distribution state-action pairs by outputting a continuous OOD score via the Morse function $M_{\phi}$.
>
> We will revise Section 3.2 to incorporate these clarifications, ensuring the motivation behind our choices is more transparent to the reader.
>
> **2. Tightness of the Approximation in Eq. (4)**
>
> This is an excellent question regarding the theoretical guarantees of our approximation. The closed-form solution in Proposition 3.1  is derived by making two simplifying assumptions, which are detailed in Appendix D : (1) we use a first-order Taylor expansion to approximate the objective function $M_{\phi}(s,a)$, and (2) we tighten the KL divergence constraint.
>
> Deriving a tight, analytical approximation bound is theoretically challenging due to the nature of these assumptions. However, we empirically analyze the impact of this approximation in our ablation study on the KL constraint parameter $\epsilon$, presented in Figure 2(c) . As we discuss on page 9 (lines 306-309) , the results confirm our theoretical intuition: performance degrades if $\epsilon$ is too large (as the first-order approximation becomes inaccurate) or too small (as exploration is overly restricted).
>
> The primary motivation for this approximation was to develop a computationally efficient method suitable for real-time application in the RL loop, avoiding the high overhead of iterative solvers at each timestep. Our experiments demonstrate that with a reasonably chosen $\epsilon$ (e.g., 0.01), this approximation is highly effective in practice. We will add a brief discussion in the main text to acknowledge this trade-off between theoretical tightness and computational feasibility.
>
> **3. Collection of the Dataset $\mathcal{D}^{cp}$**
>
> We apologize for the lack of clarity on this point. The dataset $\mathcal{D}^{cp}$ represents the **candidate pool of trajectories to be queried**. Our method is designed to be integrated with existing query selection strategies. In our experiments, we integrated PPE with the state-of-the-art QPA algorithm.
>
> Therefore, $\mathcal{D}^{cp}$ is first populated using the **policy-aligned query technique from QPA** (Hu et al., 2023), which selects trajectories relevant to the current policy. Our mixture distribution query method (Algorithm 1) then acts as a **re-sampling post-processing step** on this candidate pool $\mathcal{D}^{cp}$ to select the final trajectory pairs for which we request preference labels . We will revise the text to make this process explicit.
>
> **4. Preference Query Mechanism in Experiments**
>
> For our main benchmark experiments in Section 4.1, we followed the standard protocol used in prior PbRL literature (e.g., Christiano et al., 2017 Lee et al., 2021b). The preferences are provided by a **simulated oracle**. This oracle has access to the ground-truth returns of the trajectories. When presented with a pair of trajectory segments $(\tau^0, \tau^1)$, it provides a preference for the segment with the higher cumulative ground-truth reward. This ensures our experimental results are reproducible and allows for a fair and controlled comparison against the baseline methods. We note this practice in our motivating example on page 3 (lines 130-131).
>
> To validate our approach in a more realistic scenario, we also conducted supplementary experiments with **real human feedback**, as mentioned in Appendix E.7. We will clarify the use of a simulated oracle for the main experiments in Section 4.
>
> **5. Discussion of State-of-the-Art Baselines**
>
> Thank you for this suggestion. We agree that a more explicit high-level comparison of the baselines would strengthen the paper.
>
> **6. Evaluation Metric in Table 1**
>
> We sincerely apologize for this omission. You are correct that the metric should be clearly stated. We will revise the caption of Table 1 and the main text to clarify:
>
> * For the **DMControl** suite tasks, the reported values are the **average episode returns**.
> * For the **MetaWorld** tasks, following the evaluation protocol of previous work, the metric is the **ground truth success rate**, reported as a percentage.
>
> **7. Clarification of Claims**
>
> "Limited query budget":** In our experimental setup, the agent is allowed to query for a fixed total number of preferences throughout training. This is enforced by setting a query frequency $K$ and a total number of environment steps (as seen in Algorithm 2 and the x-axis of Figure 2. This finite number of available queries constitutes the "limited budget." Our goal is to maximize performance within this constraint. We will make this definition explicit in the experimental setup section.
>
> **8. Generality of the KL Constraint $\epsilon$**
>
> This is an important question about the practical application of our method. For all experiments reported in Table 1, we used the **fixed value of $\epsilon=0.01$ across all tasks and environments without any per-task tuning**. Our results show that PPE consistently outperforms the baselines across a diverse set of locomotion and manipulation tasks. This demonstrates that the recommended value of $\epsilon=0.01$ is robust and generalizes well, alleviating the need for costly hyperparameter tuning for each new domain, which we consider a practical strength of our approach.
>
> **9. Bolding of Results in Table 1**
>
> Thank you for your careful eye and for pointing out this inconsistency. Your point is absolutely correct; bolding should be reserved for results that are statistically significantly superior, not just numerically higher when standard deviations overlap. As mentioned in our checklist response, we have performed Welch's T-tests to assess statistical significance (Appendix E) . In the revised manuscript, we will correct Table 1 and ensure that a result is bolded only if it is statistically significant (e.g., p-value < 0.05) compared to the next-best baseline.
>
> **10. Minor Issues**
>
> We appreciate you catching these details. We will make corrections in the revised manuscript.
> ---
>
> Once again, we thank you for your thorough and valuable feedback. We are confident that addressing these points will make our paper stronger, clearer, and more impactful.
>
> Sincerely,
>
> The Authors

---

> > ### Comment · Reviewer_iwcb · 2025-08-04
> >
> > I thank the authors for the clarifications and detailed responses. I will increase my score to 4 in light of the answers to my questions. I strongly suggest that the authors incorporate these clarifications in the main text in case of acceptance.

---

> > > ### Author Response · Authors · 2025-08-04
> > >
> > > Dear Reviewer iwcb,
> > >
> > > Thank you very much for your thoughtful consideration of our rebuttal and for increasing your score.
> > >
> > > As you suggested, we will be sure to incorporate the clarifications to improve its clarity and presentation. Your guidance has been invaluable in helping us strengthen our paper.
> > >
> > > Thank you once again for your positive and helpful engagement.
> > >
> > > Sincerely,
> > >
> > > The Authors

---

### Official Review · Reviewer_1gpz · 2025-06-30

**Clarity:** 2
**Significance:** 3
**Originality:** 3
**Rating:** 4
**Confidence:** 4

**Summary:**

This paper introduces a novel shifts-aware reward learning framework named SAMBO-RL to mitigate distribution shift in model-based offline RL. Theoretical and empirical results demonstrate the effectiveness of the approach across multiple benchmarks.

**Questions:**

Terminology
- Is "preference buffer" an established term in the literature? If not, could the authors provide a clearer explanation or citation for this term?

The description of the experimental setup in Section 3.1 is vague, and some findings raise questions:
- Is it correct that the training region is identical to the evaluation region when their sizes are equal (e.g., training region = 3 and evaluation region = 3)?
- Fig. 1(b), line 146: the result that "variance in outputs from ensemble reward models does not distinguish whether the transition belongs to the training region" is surprising. How is the ensemble implemented? Were the ensemble members deliberately made diverse during training?
- Line 124: how are ground-truth returns obtained? Returns $G_t$ depend on the sequence of rewards, which in turn depend on the states the agent visits and the actions it takes. Which policy is employed in the calculation of ground-truth returns?
- Fig. 1(c): the blue curve improve as the evaluation region expands. Could the authors explain this phenomenon?
- Line 129: when the training region is small, there must be a lot of duplicated trajectories while sampling 1000 trajectories. How is this handled?
- Fig. 1(d), could the authors elaborate on how the conclusion in lines 151–153 is derived? The result is not surprising from a learning perspective, as this setup is only for in-distribution cases, and more data generally improves performance. The conclusion should clarify that more in-distribution data improves in-distribution performance.
- Appendix G contains relevant details and should be integrated into the main text for better understanding.

Methodology
- line 191, how should the "target policy" be understood in the context of the framework?
- Are the mixture ratio $k$ and KL constant $\varepsilon$ sensitive to different tasks? If so, how are they tuned for each task?
- In appendix F.1, how should the values in Table 8 be interpreted? Could the authors evaluate the method on more tasks? Is it correct that a separate Morse Network must be trained for each task? What is the type and size of the data used for training the Morse Network in the experiments?

Ablation study
- Could the authors elaborate on the ablation setups: 1) without MDQ and with EXT, and 2) with MDQ and without EXT? My understanding is that MDQ relies on EXT, and sole EXT does not directly impact training. Providing two pseudocode snippets, similar to Algorithm 1, for these setups would be helpful.

**Ethical Concerns:**

["NO or VERY MINOR ethics concerns only"]

**Final Justification:**

I raise my score from "Borderline reject" to "Borderline accept". My initial concerns have been addressed.
However, the revised manuscript still needs clearer presentation to eliminate misunderstandings I raised earlier. Additionally, the significance of the proposed method is limited by inuring additional hyper-parameters, as I pointed out in my initial comment: "The proposed method introduces additional hyper-parameters and models, which may limit its practicality and scalability in real-world applications due to the increased complexity and tuning requirements." For these reasons, I would suggest a "borderline accept" to this submission.

**Limitations:**

See weaknesses and questions.

**Quality:**

3

**Strengths And Weaknesses:**

Strengths
- The paper is well-written and easy to follow, with logical structuring and clear explanations of the methodology.
- The proposed method is novel on addressing distribution shift in PbRL.
- The paper includes sufficient technical details in both the main text and appendix, making the method comprehensible.

Weaknesses
- Some aspects of the methodology and experimental setup require further clarification. Specific details are outlined in the questions below.
- The proposed method introduces additional hyper-parameters and models, which may limit its practicality and scalability in real-world applications due to the increased complexity and tuning requirements.

---

> ### Author Rebuttal · Authors · 2025-07-31
>
> Dear Reviewer 1gpz,
>
> We would like to extend our sincere gratitude for your detailed and insightful feedback on our manuscript. Your thoughtful comments and questions have been invaluable in helping us identify areas for improvement. We are encouraged that you found our paper well-written, our method novel, and our technical details sufficient.
>
> We have carefully considered all your suggestions and provide the following point-by-point responses to your questions and concerns. We hope these clarifications will address the issues you raised and demonstrate the value of our work.
>
> #### **1. Terminology: "Preference Buffer"**
> Thank you for this question. The term "preference buffer" is a direct and intuitive adaptation of the standard "replay buffer" concept for the Preference-based RL (PbRL) setting. In PbRL, instead of storing state-action-reward-next-state tuples, the agent collects and learns from preference data. This data, consisting of pairs of trajectory segments and a human preference label, i.e., $(\tau^0, \tau^1, y_p)$, is stored in a dataset $\mathcal{D}_p$. While the exact phrase "preference buffer" may not be universally standardized, the concept of a buffer or dataset for storing preferences is fundamental to nearly all modern PbRL algorithms. Foundational works like Christiano et al. (2017) and Lee et al. (2021a, 2021b) all rely on collecting and sampling from such a dataset of preferences to train the reward model. We will clarify this in the revised text by explicitly defining it as the dataset $\mathcal{D}_p$ that stores preference tuples and citing these foundational works to ground the concept.
>
> #### **2. Experimental Setup in Section 3.1 (Motivating Example)**
>
> We apologize for the lack of clarity. The training and evaluation regions are distinct spatial areas within the 9x9 grid world, as depicted in **Figure 1(a)**. The training region is centered, while the evaluation region expands outwards from the center. Therefore, even when their side lengths are equal (e.g., both are 3x3), they are not identical unless both are the full 9x9 grid. For instance, when the training region is 3x3 and the evaluation region is 3x3, the evaluation is performed on a 3x3 area that may or may not perfectly overlap with the training region, depending on the specific setup shown in Figure 1(a). We will revise the caption and text in **Section 3.1** to make this distinction clearer.
>
> This is an excellent question that highlights a key motivation for our work. The ensemble was implemented using a standard approach: we trained multiple reward models (5 in our experiment) on the exact same preference buffer, with diversity introduced only through different random network initializations and different mini-batch shuffling during training. This is a common technique for estimating uncertainty (e.g., in RUNE). Our finding, as shown in **Figure 1(b)**, is that the variance produced by such an ensemble does not serve as a reliable indicator for distinguishing in-distribution (ID) from out-of-distribution (OOD) transitions. The variance can be high for complex ID transitions and low for simple OOD ones. This observation motivates our decision to use a dedicated OOD detection mechanism (the Morse Neural Network) rather than relying on ensemble variance for guiding exploration.
>
> The ground-truth return for a given trajectory is calculated by summing the predefined, ground-truth rewards of each cell visited in that trajectory. In this specific experiment, trajectories are generated by a random walk (i.e., uniformly sampling actions), not by a learned policy. The goal is not to evaluate a policy, but to assess the reward model's ability to correctly rank pre-existing trajectories. Thus, the ground-truth return $G(\tau)$ for a trajectory $\tau$ is simply the sum of the true rewards encountered, and no single policy is employed to generate all trajectories for evaluation.
>
> There may be a misunderstanding of the plot. In **Figure 1(c)**, the blue curve (representing a tiny 1x1 training region) shows a *sharp decrease* in performance as the evaluation region expands. It starts with a high Spearman correlation when the evaluation region is also 1x1 (i.e., evaluating on the same single cell it was trained on) and then plummets as the evaluation region grows to include states the reward model has never seen. This result strongly supports our central claim: reward models generalize poorly to OOD trajectories, and their reliability is fundamentally tied to the coverage of the preference buffer.
>
> When sampling 1,000 trajectory pairs from a small region, duplicates are indeed likely. We sampled with replacement. For this motivating experiment, the presence of duplicate trajectories in the preference buffer is not an issue. It simply means the reward model observes the same preference comparisons multiple times during training, which is standard. The key takeaway of the experiment is about the effect of the spatial *coverage* of the training data, not the diversity of trajectories within that covered region.
>
> We agree that moving key details about the motivating example from the appendix to the main text would improve readability. In the revised manuscript, we will incorporate the most relevant information from Appendix G into **Section 3.1**, space permitting.
>
> #### **3. Methodology**
>
> In our framework, the "target policy" $\pi_T$  refers to the current actor's policy that we are actively trying to improve. In our implementation, which uses SAC (as mentioned in Algorithm 2), $\pi_T$ is the current SAC policy. Our exploration policy, $\pi_E$, is derived from $\pi_T$ using the proximal-policy extension method to gather data that is most useful for the *next* update of $\pi_T$. We will add this clarification to the text.
>
> This is a crucial question regarding the practicality of our method. As shown in our ablation study in **Section 4.2** (Figures 2b and 2c), we investigated the sensitivity of these parameters. We found that performance is robust within a reasonable range of values. Optimal performance was achieved around $\kappa=0.5$  and $\epsilon=0.01$. Based on this, we used these fixed values for all tasks in our experiments without any per-task tuning. This indicates that while PPE introduces new hyperparameters, they are not overly sensitive and do not require a costly tuning process for each new task, thus supporting the practicality of our approach.
>
> * **Interpretation of Table 8:** We apologize for not having the appendix available to you. Table 8 in Appendix F.1 reports the performance of our Morse Network as an OOD detector on several tasks, using standard metrics like AUROC and AUPR. High values for these metrics indicate that the network is effective at distinguishing between in-distribution data (from the preference buffer) and out-of-distribution data, which is crucial for our method.
> * **Evaluation on More Tasks:** Yes, a separate Morse Network must be trained for each task, as the distribution of state-action pairs is task-specific.
> * **Training Data:** The Morse Network is trained using the state-action pairs $\{(s,a)\}$ available in the current preference buffer $\mathcal{D}^p$, as described by the loss function in **Eq. (2)*. The size of this buffer grows as more feedback is collected. In our experiments, the training starts with an initial set of preferences and is periodically updated using all data in $\mathcal{D}^p$. We will ensure these details are clearly stated in the appendix of the final version.
>
> #### **4. Ablation Study**
>  Your understanding is perfectly correct, and this highlights the synergy between the two components. We will gladly add pseudocode for these ablations in the appendix of the revised paper.
> * **Ablation 1 (QPA + EXT w/o MDQ):** In this setup, the agent explores using our proximal-policy extension method (**EXT**) to gather OOD data into the main replay buffer. However, for querying, it uses a standard method (in our case, the policy-aligned query from QPA) which does not explicitly prioritize selecting this newly found OOD data for labeling. As a result, the *preference buffer's* coverage does not expand effectively, and the reward model's OOD generalization does not improve, leading to minimal performance gains.
> * **Ablation 2 (QPA + MDQ w/o EXT):** Here, the agent explores normally (via the SAC policy's stochasticity), without active OOD exploration from **EXT**. The **MDQ** mechanism is then used to select pairs for querying. However, since the replay buffer contains very little actively-sought OOD data, the OOD sampling part of MDQ ($P^{out}$) has a poor set of candidates to select from. This also leads to ineffective expansion of the preference buffer's coverage.
>
> As you rightly pointed out, the two components are complementary. EXT provides the OOD data, and MDQ ensures this data is incorporated into the preference buffer to improve the reward model. This synergy is why the full **PPE** algorithm significantly outperforms either component alone, as shown empirically in **Figure 2(a)**.
>
> ***
>
> Once again, we thank you for your rigorous review and constructive suggestions. We believe that by incorporating these changes, we can significantly improve the clarity and impact of our paper.
>
> Sincerely,
> The Authors

---

> > ### Comment · Reviewer_1gpz · 2025-08-06
> > **Thank you for the indetailed clarification**
> >
> > Most of my initial concerns and questions are satisfactorily addressed. I recommend that technical details of the implementation of the ensemble-based method and more in-depth discussion regarding its performance are included in the revised version. The underperformance of this method is somewhat surprising and serves as a key motivation for your methodology.
> > Regarding Figure 1(c), it is reasonable to see the shape decrease when the evaluation region increases to 2. However, it is still unclear why there is an improvement in performance when the evaluation region increases from 2 to 7. I would appreciate it if the authors could provide further explanation for this phenomenon.

---

> > > ### Author Response · Authors · 2025-08-06
> > >
> > > Dear Reviewer 1gpz,
> > >
> > > We are glad that our initial clarifications addressed most of your concerns. Your follow-up questions astutely pinpoint the key aspects of our paper that warrant further elaboration, and we are deeply grateful for this.
> > >
> > > ### 1. In-depth Discussion on the Underperformance of the Ensemble Method
> > >
> > > We fully agree with your perspective. A thorough explanation of why the ensemble-based method underperforms in our setting is crucial for motivating the core of our work.
> > >
> > > The failure of OOD detection methods based on ensemble variance stems from the **non-uniqueness** and **scale ambiguity** of the reward function, which are inherent properties of preference-based learning.
> > >
> > > **The Nature of Preference-based Constraints — Non-uniqueness of the Reward Function:** In Preference-based Reinforcement Learning (PbRL), we learn a reward function $\hat{r}$ from preference data $(\tau^1, \tau^2, \mu)$. The preference label $\mu$ only provides an **ordinal relationship** as a constraint, i.e., $\sum \hat{r}(\tau^1) > \sum \hat{r}(\tau^2)$. It does not impose any restrictions on the specific cardinal values or the scale of the reward function. Consequently, for any given set of preference data, there exists an infinite number of valid reward functions. If $\hat{r}$ is a valid reward function, then any function $f(\hat{r})$ resulting from a strictly monotonically increasing transformation $f(\cdot)$ (e.g., $2\hat{r}$, $\hat{r}+100$, or $e^{\hat{r}}$) is also a completely valid reward function, as it yields the exact same preference judgments.
> > >
> > > During training, ensemble members converge to different but equally valid reward functions with arbitrary scales and shifts. Consequently, the resulting variance reflects these numerical differences, not true epistemic uncertainty. For example:
> > >
> > > - **On OOD data:** Models may "consistently err," yielding deceptively **low variance**.
> > > - **On complex ID data:** Models may disagree on numerical values while agreeing on the ranking, yielding misleadingly **high variance**.
> > >
> > > Therefore, in the context of PbRL, ensemble variance is not a reliable metric for OOD uncertainty. It is more likely to reflect the differences in numerical scales among model members rather than true epistemic uncertainty. This is precisely the phenomenon illustrated in Figure 1(b) and the fundamental reason we propose using a method like the Morse Network, which directly models the data distribution. Our approach does not rely on byproducts of the reward function's output (like variance) but instead trains a model to directly distinguish between in-distribution and out-of-distribution state-action pairs, providing a more direct and robust signal for exploration.
> > >
> > > We thank you again for this question, as it has prompted us to articulate this core motivation more clearly.
> > >
> > > ### 2. Explanation of the Spearman Correlation Coefficient's Behavior in Figure 1(c)
> > >
> > > Your observation regarding Figure 1(c) is very astute, and this phenomenon certainly deserves a deeper explanation. The non-monotonic behavior of the Spearman Correlation Coefficient is rooted in the interplay between the **properties of the Spearman coefficient itself** and the **reward model's generalization error in OOD regions.**
> > >
> > > Our key conclusion is that **the upward trend in the curve does not represent a genuine "improvement" in model performance.** Rather, it is an artifact of the metric's properties, reflecting a transition in the model's error pattern from "strongly anti-correlated" to "weakly correlated (tending towards random) over a larger range."
> > >
> > > The Spearman coefficient measures the strength of the **rank correlation**, or monotonic relationship, between two variables (in this case, the ground-truth returns and the predicted returns). Therefore:
> > >
> > > * **Sharp Drop (coefficient below -0.6):** This correctly demonstrates the model's catastrophic failure on OOD data, where its predictions become strongly anti-correlated with the ground truth. This is the key finding.
> > >
> > > * **Slight Rebound (from -0.6 to around 0.2):** This seemingly counter-intuitive rebound can be understood as follows: As the evaluation region continues to expand, the range of ground-truth returns of the trajectories in the pool also becomes wider. This means more pairs of "extremely good" and "extremely bad" trajectories appear, which have very large discrepancies. Even our poorly generalized reward model might be able to correctly rank these "obvious" pairs. As the evaluation area grows, the proportion of such "easy" pairs within the entire set of evaluation samples may slightly increase, thus statistically pulling up the overall Spearman coefficient from its nadir.
> > >
> > > Once again, we thank you for your invaluable feedback, which will undoubtedly make our paper more rigorous and clear.

---

> > > > ### Comment · Reviewer_1gpz · 2025-08-08
> > > > **Thanks for the further clarification**
> > > >
> > > > Your intuitive explanation of the upward trend is reasonable. While it does not undermine the core motivation of the proposed method, clarifying this phenomenon would help readers better understand the behavior of the Spearman coefficient and the task setup. I suggest adding the explanation you provided in the appendix.

---

> ### Author Response · Authors · 2025-08-09
>
> Dear Reviewer 1gpz,
>
> We would like to express our sincere gratitude once more for the incredibly insightful discussion. Your detailed feedback has been truly invaluable in helping us strengthen the paper.
>
> As the final decision phase approaches, we just wanted to gently inquire if, in light of our fruitful conversation, you might perhaps be willing to reconsider your final evaluation score.
>
> We deeply appreciate the significant time and effort you have already dedicated to our work. Thank you for your consideration.
>
> Sincerely,
> The Authors

---

### Official Review · Reviewer_X7X4 · 2025-07-01

**Clarity:** 2
**Significance:** 3
**Originality:** 2
**Rating:** 4
**Confidence:** 4

**Summary:**

The paper proposes the proximal policy exploration scheme that balances explicit exploration of out-of-distribution and in-distribution trajectories via Morse network-based uncertainty estimation. The authors argue that generalization of reward models is limited due to insufficient preference buffer coverage, and demonstrate this in a motivating example. The use of the Morse network as an OOD detector is taken from existing work, and similarly the exploration scheme itself is derived from the TD3-BST approach (Srinivasan & knottenbelt, 2024). They showcase their approach PPE on a series of continuous control tasks (DMControl & MetaWorld) with simulated preference feedback, and provide additional results in the supplementary material.

**Questions:**

- How many participants were recruited in the human feedback experiments? How many queries were used? Can you provide additional information about the experiment? In theory, this would be a very insightful and important experiment, but a lot of detail is missing
- How does PPE compare to "standard" SAC? I know that SAC baselines have been reported in previous work, but i would appreciate it being mentioned at least in the appendix for the environments and hyperparamter settings reported in this paper

**Ethical Concerns:**

["NO or VERY MINOR ethics concerns only"]

**Final Justification:**

The presented algorithm has clear merits, and the paper is well-crafted overall. However, there remain some limitations in terms of evaluation, which makes it difficult to fully assess the effectiveness of the method.

**Limitations:**

Limitations are not sufficiently discussed, e.g.
- choice of hyperparameters, stability
- experiments should be extended to other environments and tasks beyond control benchmarks
- larger scale investigation of human preference feedback

**Paper Formatting Concerns:**

No concerns

**Quality:**

3

**Strengths And Weaknesses:**

Strengths:
- As mentioned, the exploration-exploitation trade-off remains the fundamental challenge in RL, and so i think this research is highly valuable and potentially high impact
- The reported empirical results are impressive, outperforming competing methods
- The appendices are very comprehensive, and provide ample detail. I particularly like the inclusion of a justification for using the Morse network
- Ablation studies for the effect of different algorithmic components are highly appreciated (Section 4.2)
- The proposed algorithm seems widely applicable if Morse networks are scalable, the authors also investigate the integration of PPE into other base algorithms

Weaknesses:
- Different contributions seem intermingled in the algorithm: There is the uncertainty estimation via Morse networks (which is not really evaluated except for using the OOD detection performance in isolation), the exploration mechanism, and the query selection. Therefore, i think proper ablation studies are important. However, i find them insufficient for now. Using just a single environment (especially since the results seem to have large error bars) limits their significance. I would appreciate an extension of the experiments in Figure 2a)
- As the topic of exploration is very prevalent in RL, i think the related work should discuss existing approaches (including the mentioned baselines) in more detail.
- For me, this paper misses certain basic ablations as a sanity check (e.g. varying the entropy bonus coefficient, which is fine-tuned for PPE)
- The motivating example is not very convincing for me. The experiment itself is lacking detail (i.e. the network architecture, training parameters, etc. are not described), and the outcome is pretty straight-forward, i.e., fact that ML models do not automatically extrapolate is clearly established
- Source code is not provided

While i see some flaws right now, i think the overall algorithm and results are fair, and i think many of my concerns could be addressed as part of a revision.

---

> ### Author Rebuttal · Authors · 2025-07-31
>
> Dear Reviewer X7X4,
>
> We would like to express our sincere gratitude for your time and effort in providing such a detailed and constructive review of our manuscript. Your insightful comments and suggestions are invaluable to us, and they have helped us identify key areas for improvement. We are encouraged that you find our research on the exploration-exploitation trade-off to be valuable and potentially high-impact, and that you appreciate our empirical results and comprehensive appendices.
>
> We have carefully considered all your feedback and provide a point-by-point response below. We believe that by addressing these points, we can significantly strengthen our paper.
>
> Responses to Weaknesses
> 1. Regarding the sufficiency of ablation studies:
>
> We agree with you that proper ablation studies are crucial for disentangling the contributions of the different components of our proposed algorithm. We appreciate you acknowledging the importance of the ablation study presented in Section 4.2.
>
> As you noted, the current ablation in Figure 2(a) was conducted on the Walker-walk task. Our rationale was to use this as a representative environment to demonstrate the synergy between the proximal-policy extension method (EXT) and the mixture distribution query method (MDQ). The results clearly show that using either component in isolation does not lead to significant improvements, and their complementary nature is what drives the superior performance of the full PPE algorithm.
>
> We take your point about the statistical significance and the desire for broader evidence. In the revised version of our manuscript, we will expand our ablation studies to include additional environments. This will provide more robust evidence for our claims and better illustrate the individual contributions of each component of PPE.
>
> 2. Regarding the level of detail in the Related Work section:
>
> Thank you for your suggestion to expand the discussion on existing exploration approaches. In Section 5, under "Exploration in RL", we do provide a brief overview, noting that prior PbRL work has focused on reward model disagreement for exploration, while our approach focuses on maximizing preference buffer coverage. We also situate our work within the broader context of RL exploration strategies like uncertainty-driven and information-theoretic methods.
>
> However, we agree that a more detailed comparison with the exploration strategies used in our baseline methods would strengthen the paper.
>
> 3. Regarding missing basic ablations (e.g., entropy bonus):
>
> This is an excellent point. To ensure a fair comparison in our main experiments, we utilized the official code repositories of the baseline algorithms and their recommended hyperparameter settings wherever possible. Our method, PPE, is integrated into the QPA framework, and the underlying RL algorithm is SAC. The hyperparameters for SAC were kept consistent with the baseline to isolate the effects of our proposed exploration and query methods.
>
> While we did not perform a specific ablation on the SAC entropy bonus coefficient, we did conduct sensitivity analyses on the key new hyperparameters introduced by PPE, namely the mixture ratio $\kappa$ and the KL constraint $\epsilon$. We agree that an ablation on the entropy coefficient would be informative. ]
>
> 4. Regarding the motivating example:
>
> We appreciate your critical feedback on the motivating example in Section 3.1. While we acknowledge that the general principle of neural networks struggling with out-of-distribution (OOD) data is well-established, our goal with this experiment was more specific to the PbRL context. We aimed to systematically demonstrate that this "distributional mismatch" is a critical issue where reward models trained on static buffers fail on OOD trajectories from policy exploration, and that "such failures in policy-proximal regions directly misguide iterative policy updates".
>
> The key takeaways we wanted to highlight were:
> • The necessity of making preference buffer coverage an explicit optimization objective, which is not standard practice.
> • A non-trivial finding that a common method for uncertainty estimation, ensemble variance, fails to distinguish between in-distribution and OOD transitions in this setting, as shown in Figure 1(b).
> • The crucial insight from Figure 1(d) that simply exploring OOD data is not enough; a balance between in-distribution and OOD feedback is essential for maintaining the reward model’s accuracy.
>
> To address your concern about the lack of detail, we will add the full experimental parameters for this example (e.g., network architecture, training details) to the appendix in our revision to improve clarity and reproducibility.
>
> 5. Regarding the availability of source code:
>
> We apologize for not providing the code with the initial submission. As stated in our checklist, we are fully committed to open science and will make the complete source code publicly available upon acceptance of the paper.
>
> Responses to Questions
> 1. Regarding the details of the human feedback experiments:
>
> Thank you for your interest in this aspect of our work. The primary quantitative results in our paper are based on experiments with a simulated oracle to ensure controlled and extensive comparisons, as is common practice in the field. However, we agree that validation with real human feedback is very important.
>
> As mentioned in the checklist, we did conduct experiments with real human participants. We recruited volunteers who were compensated for their time to provide preference labels. The full instructions provided to these participants are included in the supplementary material, and Appendix E.7 details how this feedback was incorporated. The results are provided as qualitative evidence in the form of videos of learned agent behaviors. We acknowledge that quantitative details such as the number of participants and queries were not in the main text. We will add a detailed description of the human feedback study, including the number of participants, the number of queries collected per participant, and other relevant experimental details, to the appendix in the revised version.
>
> 2. Regarding the comparison to a standard SAC baseline:
>
> The goal is not to outperform SAC trained with the ground-truth reward, but rather to learn a policy that is as close as possible to that upper bound while using a minimal amount of preference feedback. Previous work in PbRL has established the performance of SAC with ground-truth rewards on these benchmarks. However, for completeness and to provide a clear performance ceiling for our experimental setting, we agree that including this baseline is beneficial.
>
> Responses to Limitations
>
> We thank you for pushing us to expand our discussion on the limitations of our work. This is crucial for positioning our contributions accurately. The current discussion in Section 6 acknowledges that our query method could be improved by considering the information content of behavior pairs. We will expand this section in our revision to incorporate the excellent points you raised.
>
> Once again, we thank you for your detailed and insightful review. We are confident that by incorporating your suggestions, we can substantially improve the quality and clarity of our paper.
>
> Sincerely,
>
> The Authors

---

> > ### Comment · Reviewer_X7X4 · 2025-08-01
> >
> > Thank you for your response. I have already given a very positive score and I will stick to it.

---

> > > ### Author Response · Authors · 2025-08-04
> > >
> > > Dear Reviewer X7X4,
> > >
> > > Thank you for your quick response and for maintaining your positive score.
> > >
> > > We would also be very grateful for any opportunity for further discussion, especially if it might help to further raising the score of our work. We are ready and willing to provide any additional clarifications.
> > >
> > > Thank you again for your time and thoughtful comment!
> > >
> > > Sincerely,
> > >
> > > The Authors

---

> > > > ### Comment · Reviewer_X7X4 · 2025-08-05
> > > >
> > > > Thanks for your response. I would advise the authors to add some additional ablations. However, as the outcome of the additional experiments is unclear at the current time, and necessary additions to the paper would amount to significant changes, i am not ready to raise my score further. Still, as part of the discussion, i am willing to advocate for acceptance.

---

### Official Review · Reviewer_Eg1k · 2025-07-03

**Clarity:** 3
**Significance:** 2
**Originality:** 2
**Rating:** 4
**Confidence:** 3

**Summary:**

This paper addresses a critical challenge in Preference-Based Reinforcement Learning (PbRL): the limited generalization of reward models trained on preference data with insufficient coverage. The core contribution is a novel algorithm, Proximal Policy Exploration (PPE), designed to explicitly and actively expand the coverage of the preference buffer. PPE consists of two main components: 1) a "proximal-policy extension" method that uses a Morse Neural Network to detect out-of-distribution (OOD) states and guides the policy to explore these novel regions under a KL-divergence constraint, and 2) a "mixture distribution query" method that balances the sampling of newly discovered OOD data and familiar in-distribution data for human labeling. This dual approach aims to improve the reward model's global accuracy without sacrificing its stability, ultimately leading to better policy performance. The authors validate their method on a suite of continuous control benchmarks, demonstrating performance gains over existing state-of-the-art PbRL algorithms.

**Questions:**

See weakness

**Ethical Concerns:**

["NO or VERY MINOR ethics concerns only"]

**Final Justification:**

Some concerns have been addressed. But I doubt the method can extend to LLMs. It introduces an additional network, making scaling harder for already large models, and perturbing the logit space risks excessive, unproductive exploration. In games or robotics, such inefficiency may be tolerable, but for LLMs it often produces meaningless outputs that offer no training benefit. As related works have not evaluated on LLMs, I consider this concern to be of lower priority. So I maintain the score to 4 (weak accept).

**Limitations:**

yes

**Quality:**

3

**Strengths And Weaknesses:**

### Strength

Well-motivated:  Through an intuitive "motivating example," the authors identify two core challenges in PbRL: 1) the reward model needs to explore out-of-distribution (OOD) regions to expand its cognitive boundaries and improve generalization, and 2) simultaneously, the model must continuously review in-distribution data to prevent performance degradation on known regions. The proposed PPE algorithm, with its two core components (proximal-policy extension and mixture distribution query), solves these two key problems.

Strong Ablation Studies: The paper's ablation studies are very thorough and convincing. The results clearly show that removing either component of PPE leads to a significant drop in performance. This proves that both problems initially identified by the authors are crucial, which in turn validates the correctness of their core motivation and the completeness of their method's design.

High Clarity: The paper's organization is clear, the writing is fluent. From the problem statement to the method's description and the experimental validation, the entire process is coherent.

### Weaknesses

1.  Scope of OOD Detection: The OOD detection mechanism in Equation 2 appears to primarily focus on whether the action `a` is novel for a given state `s`. It is unclear how the novelty of the state `s` itself is factored in. If the agent encounters a completely new state but performs a very familiar action, the current method might fail to identify this transition as an OOD event that warrants exploration.

2.  Unclear Objective of the Loss Function: The paper interprets the loss function in Equation 2 as minimizing a KL divergence. This is confusing, as $M_\phi$ is not defined as a probability distribution. The form of Equation 2 strongly resembles a contrastive learning objective, such as the InfoNCE loss [1]. The paper would be much clearer if the authors either elaborated on the connection between their loss function and the contrastive learning framework or provided a more detailed explanation of their KL divergence objective.

3.  Applicability to Language Models: The paper's experiments are focused entirely on continuous control tasks. Given the immense potential of preference-based learning for aligning large language models (LLMs), a natural question is whether the "coverage-driven" approach proposed here could be extended to the language domain. Discussing how this method might be adapted to discrete and high-dimensional spaces like text generation would broaden the impact of this work.

[1] Understanding Contrastive Representation Learning through Alignment and Uniformity on the Hypersphere. ICML 2021.

---

> ### Author Rebuttal · Authors · 2025-07-31
>
> Dear Reviewer Eg1k,
>
> Thank you for your thorough review and insightful feedback on our manuscript. We are grateful for your positive comments on the motivation, ablation studies, and clarity of our work. Your constructive criticisms are invaluable, and we believe that addressing them will significantly improve the quality of our paper.
>
> Below, we address each of the weaknesses you identified in a point-by-point manner.
>
> ---
>
> ### **Response to Weaknesses**
>
> **1. Weakness: Scope of OOD Detection**
>
> Thank you for this insightful question. We apologize if this aspect was not made sufficiently clear in the manuscript. We would like to clarify that our out-of-distribution (OOD) detection mechanism does, in fact, account for the novelty of the state `s`.
>
> The core of our OOD detection is the perturbation model $\hat{a} = f_{\phi}(s, a)$, which is a neural network that takes both the current state `s` and action `a` as input. The OOD score is then derived from the difference between the perturbed action $\hat{a}$ and the original action `a`, specifically $M_{\phi}(s,a)=1-K_{RBF}(f_{\phi}(s,a),a)$.
>
> Because $f_{\phi}$ is a function of `s`, its output is conditioned on the state. If the agent encounters a state `s` that is OOD with respect to the states present in the preference buffer $\mathcal{D}^p$, the neural network $f_{\phi}$ is expected to generalize poorly. Consequently, it will likely produce a significant perturbation, meaning $f_{\phi}(s, a)$ will be far from `a`, even if the action `a` itself is a common one. This will result in a high OOD score $M_{\phi}(s,a)$, correctly identifying the transition $(s, a)$ as novel and warranting exploration.
>
> In our revision, we will add a more detailed explanation in Section 3.2 to explicitly state that the novelty of the state `s` is implicitly captured by the perturbation model $f_{\phi}(s, a)$, thereby ensuring the OOD detection mechanism is sensitive to novel states as well as novel actions.
>
> ---
>
> **2. Weakness: Unclear Objective of the Loss Function**
>
> Thank you for this very sharp observation and for suggesting the connection to contrastive learning. We agree that our explanation was insufficient and could lead to confusion.
>
> The formulation of our loss function in Equation 2 is directly adopted from the work on Morse Neural Networks by Dherin et al. (2023) . As stated in our paper, the objective is to minimize the KL divergence *between unnormalized measures*, i.e., $D_{KL}(\mathcal{D}^{p}(s,a)||1-M_{\phi}(s,a))$, based on principles from Information Geometry (Amari, 2016). This is distinct from the standard KL divergence between probability distributions, and we apologize for not making this critical distinction clear.
>
> Your insight regarding its resemblance to contrastive learning is excellent and provides a very helpful intuition. Indeed, the loss function operates on similar principles:
> * The first term, $\frac{\lambda^{2}}{2}||f_{\phi}(s,a)-a||^{2}$, encourages the model to produce minimal perturbation for in-distribution pairs from $\mathcal{D}^p$, effectively aligning the representation with the target (akin to positive pairs).
> * The second term, involving a sum over uniformly sampled actions $a_u$, pushes the model's output for these "negative" pairs away from their corresponding actions, promoting uniformity over the action space for a given state.
>
> This "alignment and uniformity" characteristic is the hallmark of contrastive learning.
> ---
>
> **3. Weakness: Applicability to Language Models**
>
>
> Thank you for raising this excellent point. We agree that extending our coverage-driven approach to the alignment of Large Language Models (LLMs) is a highly promising and important future direction. While our current experiments are in continuous control, the underlying principles of our method can be adapted to discrete domains like language.
>
> Here is a potential path for adaptation:
> * **OOD Detection for Language:** The state `s` would be the context (prompt and previously generated text), and the action `a` would be the next token. Our OOD detection mechanism, based on Morse Neural Networks, could operate on the continuous embedding space of these `(context, token)` pairs. The model $f_{\phi}$ would be trained to identify generated tokens that are semantically novel or unusual given the context, relative to the examples in the preference buffer.
> * **Exploration in Discrete Spaces:** We acknowledge that our proximal-policy extension method (Equation 4), which relies on the gradient $\nabla_{a}M_{\phi}(s,a)$, is not directly applicable to the discrete token space. However, this can be addressed in several ways:
>     1.  **Exploration in Logit Space:** Exploration could be performed by adding a calculated perturbation to the pre-softmax logits, which exist in a continuous space. The OOD score $M_{\phi}$ could guide the direction of this perturbation.
>     2.  **Modulating Sampling Probabilities:** The OOD score $M_{\phi}(s, a)$ for each potential next token `a` could be used to directly modulate the final sampling probability distribution. For instance, we could up-weight the probabilities of tokens that lead to unexplored, OOD regions, thereby encouraging the LLM to generate more diverse and novel text for subsequent preference labeling.
>
> We are very optimistic about this direction. In the "Conclusion and Discussion" section of our revised manuscript [see Section 6], we will add a detailed discussion on these potential adaptations for applying PPE to LLMs, explicitly mentioning the challenges and proposing these concrete solutions.
>
> ---
>
> Once again, we thank you for your valuable and constructive feedback. We are confident that by incorporating these revisions, we can significantly strengthen our paper and address your concerns. We hope that our revised manuscript will meet the standards for acceptance.
>
> Sincerely,
>
> The Authors

---

> > ### Comment · Reviewer_Eg1k · 2025-08-01
> >
> > Thank you for the detailed response. Since I have already given a positive score, I will maintain it.

---

> > > ### Author Response · Authors · 2025-08-04
> > >
> > > Dear Reviewer Eg1k,
> > >
> > > Thank you for your prompt reply and for confirming your positive evaluation of our work!
> > >
> > > Should you have any further thoughts or questions during the discussion period, we would be delighted to address them.
> > >
> > > Sincerely,
> > >
> > > The Authors

---

### Author Response · Authors · 2025-08-07
**General Response**

Dear Area Chair and Reviewers,

We would like to extend our sincerest gratitude to  the Area Chair and all reviewers for their time and for providing insightful feedback with their expertise. **The constructive reviews have been instrumental in helping us identify areas to improve the clarity and impact of our paper**.

We appreciate the reviewers' positive recognition of our work's motivation (Eg1k, X7X4, 1gpz), novelty and impact (1gpz, X7X4), and empirical results (Eg1k, X7X4, iwcb).

The discussion period was also highly productive, allowing us to address important questions regarding our methodology, experimental scope, and design choices. **As discussed, we will incorporate all the key clarifications and additional details into the revised version of our manuscript**. This includes expanded discussions on OOD detection and uncertainty, further details for reproducibility, and a clearer articulation of our method's broader applicability.

We are greatly encouraged by the positive reception to our rebuttal from all reviewers. Specifically, we are pleased that Reviewers Eg1k and X7X4 have confirmed they will maintain their positive scores, that Reviewer 1gpz's concerns were satisfactorily addressed, and that Reviewer iwcb has indicated they will raise their score.

Thank you once again for your expert guidance. The review process has significantly strengthened our work!

Best regards,

The Authors

---

### Decision · Program_Chairs · 2025-09-17

**Decision:**

Accept (poster)

**Comment:**

The paper introduces PPE, a coverage-driven exploration and querying scheme for preference-based RL that targets policy-proximal OOD regions (via a Morse NN detector) and balances ID/OOD querying. On DMControl and MetaWorld, PPE improves over strong baselines (notably QPA), with ablations supporting the synergy of its two components.

The acceptance is contingent on the following camera-ready updates:
* Consolidate clarifications in main text: define “preference buffer”; detail motivating setup; explain ensemble baseline implementation and Figure 1(c) behavior; specify evaluation metrics; add human-feedback details.
* Strengthen related work: contrast PPE with PEBBLE, SURF, RUNE, QPA, and exploration literature; justify Morse NN choice and Eq. 2’s information-geometric loss.
* Ablations and reporting: extend component ablations to ≥1 more environment; include entropy-coefficient sensitivity; ensure bolding only for statistically significant wins; add learning curves where helpful.
* Reproducibility: release code upon publication; document hyperparameters; note fixed KL value and mixture ratio used across tasks.